# Architecture of the baculovirus nucleocapsid revealed by cryo-EM

Xudong Jia [1,4], Yuanzhu Gao[1,2,4], Yuxuan Huang [1], Linjun Sun [1], Siduo Li [1], Hongmei Li[1], Xueqing Zhang[1], Yinyin Li[1], Jian He[1], Wenbi Wu[1], Harikanth Venkannagari [3], Kai Yang[1], Matthew L. Baker [3] ✉ & Qinfen Zhang [1] ✉

Baculovirus *Autographa californica* multiple nucleopolyhedrovirus (AcMNPV) has been widely used as a bioinsecticide and a protein expression vector. Despite their importance, very little is known about the structure of most baculovirus proteins. Here, we show a 3.2 Å resolution structure of helical cylindrical body of the AcMNPV nucleocapsid, composed of VP39, as well as 4.3 Å resolution structures of both the head and the base of the nucleocapsid composed of over 100 protein subunits. AcMNPV VP39 demonstrates some features of the HK97-like fold and utilizes disulfide-bonds and a set of inter-actions at its C-termini to mediate nucleocapsid assembly and stability. At both ends of the nucleocapsid, the VP39 cylinder is constricted by an outer shell ring composed of proteins AC104, AC142 and AC109. AC101(BV/ODV-C42) and AC144(ODV-EC27) form a C14 symmetric inner layer at both capsid head and base. In the base, these proteins interact with a 7-fold symmetric capsid plug, while a portal-like structure is seen in the central portion of head. Additionally, we propose an application of AlphaFold2 for model building in intermediate resolution density.

Baculoviridae, comprised of four known genera and about one hundred species, is a family of large, insect-specific viruses with circular double-stranded DNA (dsDNA) ranging from 80 to 180 kbp[1]. As such, baculoviruses play an important role in the suppression of insects in various ecosystems, and have been extensively applied as biopesticides for insect control[2]. Moreover, baculovirus-derived vectors have been used extensively to produce recombinant eukaryotic proteins in insect cell culture; recombinant proteins from baculoviruses are essential tools in biomedical research and vaccine development[3–5]. As foreign proteins can be displayed on the surface of baculovirus particles[6,7], they have also become an attractive new tool for gene therapy[3,7,8].

*Autographa californica* multiple nucleopolyhedrovirus (AcMNPV) is the most well studied baculovirus and was the first baculovirus to be

completely sequenced[9]. The virus was originally isolated from *Autographa californica* and contains a 134kbp circular, dsDNA genome. The DNA genome, along with the baculovirus basic DNA-binding protein P6.9 protein, is encapsulated by an enveloped, rod-shaped nucleocapsid; the major capsid protein VP39 and some additional minor proteins are known to comprise the nucleocapsid[2].

The AcMNPV life cycle has two virion phenotypes: occlusion-derived virions (ODVs) and budded virions (BVs). ODVs are responsible for the primary infection of a host, while BVs are released from the infected host cells to carry out secondary infection. Although BVs and ODVs both have an envelope surrounding a viral nucleocapsid; the shape and composition of their viral envelope are unique and distinct among the two classes of virions, BVs and ODVs share an identical nucleocapsid structure[2]. Besides the major capsid protein, VP39, which

[1]State key laboratory of biocontrol, School of Life Sciences, Sun Yat-sen University, 510275 Guangzhou, China. [2]Cryo-EM Facility Center, Southern University of Science and Technology, Shenzhen, China. [3]Department of Biochemistry and Molecular Biology, McGovern Medical School at the University of Texas Health Science Center, Houston, TX 77030, USA. [4]These authors contributed equally: Xudong Jia, Yuanzhu Gao. ✉e-mail: Matthew.L.Baker@uth.tmc.edu; Lsszqf@mail.sysu.edu.cn

forms the long, helical cylindrical body of the virions, an increasing number of structural proteins have been identified, including but not limited to P6.9[10], 38K[11,12], EC27[13], 49K[13], VP80[14,15], AC109[16–20], BV/ODV-C42[13], VP1054[21], VLF-1[22,23], and P78/83[24,25]. However, only four baculoviruses proteins - the envelopment protein GP64[26], polyhedrin[27], a sulfhydryl oxidase (P33)[28], and *Per Os* infectivity factor 5 (PIF5)[29] - have structures solved by X-ray crystallography.

Early electron microscopy imaging using negative staining and ultrathin sections of baculoviruses indicated that the virus had a periodic lattice structure of stacked rings spaced 4.5 nm apart[30]. The cylindrical nucleocapsid structure was initially reported to be composed of a 12-start helix system of monomers[31] with an apical cap on the top of the cylindrical capsid and a base structure at the bottom of the nucleocapsid cylinder[2,32]. Cryo-electron microscopy (cryo-EM) imaging of baculoviruses later revealed BVs had a remarkably elongated, ovoid shape with a large lateral space between the nucleocapsid and envelope[33]. However, no high-resolution structure of whole nucleocapsid currently exists, and thus, the major capsid protein structure, the protein components and architecture of the nucleocapsid, as well as the mechanism of nucleocapsid assembly and genome packaging, remains unknown.

In this work, to elucidate the baculovirus nucleocapsid structure and address the underlying mechanisms of genome packaging and capsid assembly, we use single-particle cryo-EM to image the nucleocapsids of AcMNPV ODVs. We not only present the atomic structure of the AcMNPV helical cylinder, but also unveil a complex assembly of six additional structural proteins, AC98, AC101(BV/ODV-C42), AC104, AC109, AC142 and AC144(ODV-EC27), at both ends of the capsid. The structural details found in AcMNPV ODVs provide a high-resolution view of baculoviruses and offer insights into the mechanism of nucleocapsid assembly.

## Results

### Architecture of the AcMNPV ODV nucleocapsid

Cryo-EM images of ODV particles (Fig. 1a, Supplementary Fig. 1a) were collected and analyzed using single-particle methods. Overall, the capsid of AcMNPV displays a "rocket-like" shape with complex symmetry (Fig. 1b–d), rather than a simple "rod" shape observed in earlier negative stain[30] or ultrathin sections[32]. Ignoring the specialized structures at both ends and only focusing on the constant diameter cylinder[34] which accounts for most of the capsid, we were able to extract ~184,000 overlapping helical trunk segments from ~1,000 micrographs, resulting in a 3.9 Å resolution helical structure for the central cylinder of the nucleocapsid (Fig. 1c, Supplementary Fig. 1b-c). To decipher the components and architecture at the nucleocapsid ends (an apical end containing the head and cap domain, and a bottom end containing the base domain), we further extracted ~24,000 sub-particles, containing both ends of the nucleocapsid from ~2000 micrographs. By applying symmetry relaxation and classification methods, we were able to reconstruct the hetero-symmetric elements of the apical and base domains (Supplementary Fig. 1b). This resulted in three structures: 1) a 14-fold-symmetric (C14) apical end, which includes a cap and head at the top end of cylinder at ~7.4 Å resolution (Fig. 1b and Fig. 2) the capsid base domain with C14 symmetry at ~7.0 Å resolution (Fig. 1d and Fig. 3) a 7-fold-symmetric (C7) "plug" in base domain also at ~7.0 Å resolution (Fig. 1d, Supplementary Fig. 1b). A composite view of the helical cylinder, head domain and base domain reconstructions is shown in Fig. 1 (Fig. 1b–d). Further, we conducted local refinements for each element of the maps and generated maps for the helical cylinder nucleocapsid protein (VP39) at 3.2 Å resolution, a 4.3 Å resolution map containing the elements in common at both ends of the nucleocapsid, a 4.8 Å resolution map of the base with an additional density (later identified as AC98), and a 4.9 Å resolution map of the base plug (Supplementary Fig. 1c).

From the density map, the major capsid protein, VP39, can be seen arranged as a dimer with helical symmetry (Fig. 2a, Supplementary Fig. 2a–b), forming a 250 Å radius (Supplementary Fig. 2c) helical cylinder. Asymmetric units, which contain two VP39s, are related to each other by a rotation of ~18.5° about the helical axis and a rise of 44.1 Å along the helical axis, generating a right-handed, 14-start helix (Fig. 1c, Supplementary Fig. 2a–b). Inside the helical nucleocapsid, seven layers of dsDNA strand are discernable (Supplementary Fig. 2c). Unexpectedly, imaged AcMNPV ODV helical cylinders varied in total length with two primary size distributions; one peak is at ~ 210 ± 12.5 nm (~20% particles), and another peak is at ~ 310 ± 12.5 nm (~ 65% particles) (Supplementary Fig. 2d).

Initially, differentiation of the head from the base in each particle from the raw micrographs was not possible, and as such, we extracted both end domains as "sub-particles". After several rounds of two-dimensional (2-D) classification, together with three-dimensional (3-D) classification and sub-particle reconstruction (Supplementary Fig. 1b), distinct features for the apical and base regions could be clearly identified (Fig. 1b, Fig. 1d, and Supplementary Fig. 2e, f). Based on structural features, the apical region could be further divided into a "cap" and a "head" (Fig. 1b, Supplementary Fig. 2f).

Surprisingly, the C14 symmetric base had nearly identical features to that of the apical head (Fig. 1b, Fig. 1d, and Supplementary Fig. 2e–f), except for that in the central regions. Both the head and base could be further segmented into three parts: an outer shell, an inner layer and a central region (Supplementary Fig. 2e, f). One additional density can be seen in the inner layer of the base when compared to the head (Supplementary Fig. 2e). Near the base and head domains, the helical cylinder constricts, while the structural elements in the base and head protrude from the outer surface of the VP39 cylinder. The head and base domains appear to terminate the helical symmetry by "squeezing" the helical cylinder at both ends, reducing the cylinder radius to ~ 200 Å.

The most significant differences between the head and base occur in their central region. In the base there is a "plug" that sits in the

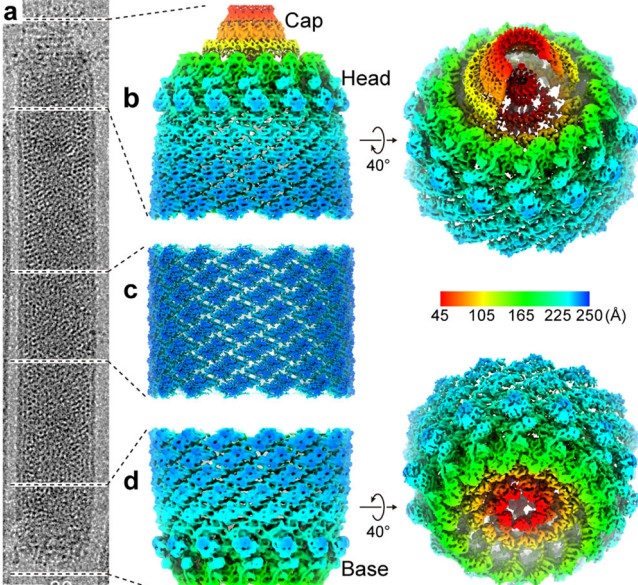

**Fig. 1 | Overall structure of *Autographa californica* multiple nucleopolyhedrovirus (AcMNPV) nucleocapsid. a** Typical image of an AcMNPV ODV nucleocapsid from two thousand cryoEM micrographs. **b** The three-dimensional (3-D) reconstruction of the apical domain, which includes the cap and head with 14-fold (C14) symmetry, is shown. On the right, a tilted view of the head shows further detail. **c** A typical portion of the capsid cylinder is shown. The helical cylinder accounts for most of the capsid which includes the regions between (**b** and **c**), and (**c** and **d**). **d** In the base, the outer portion is organized with C14 symmetry, while the middle plug (red) is arranged with C7 symmetry. On the right, a tilted view of the head shows further detail. For (**b**–**d**), the 3-D reconstruction is shown radially colored.

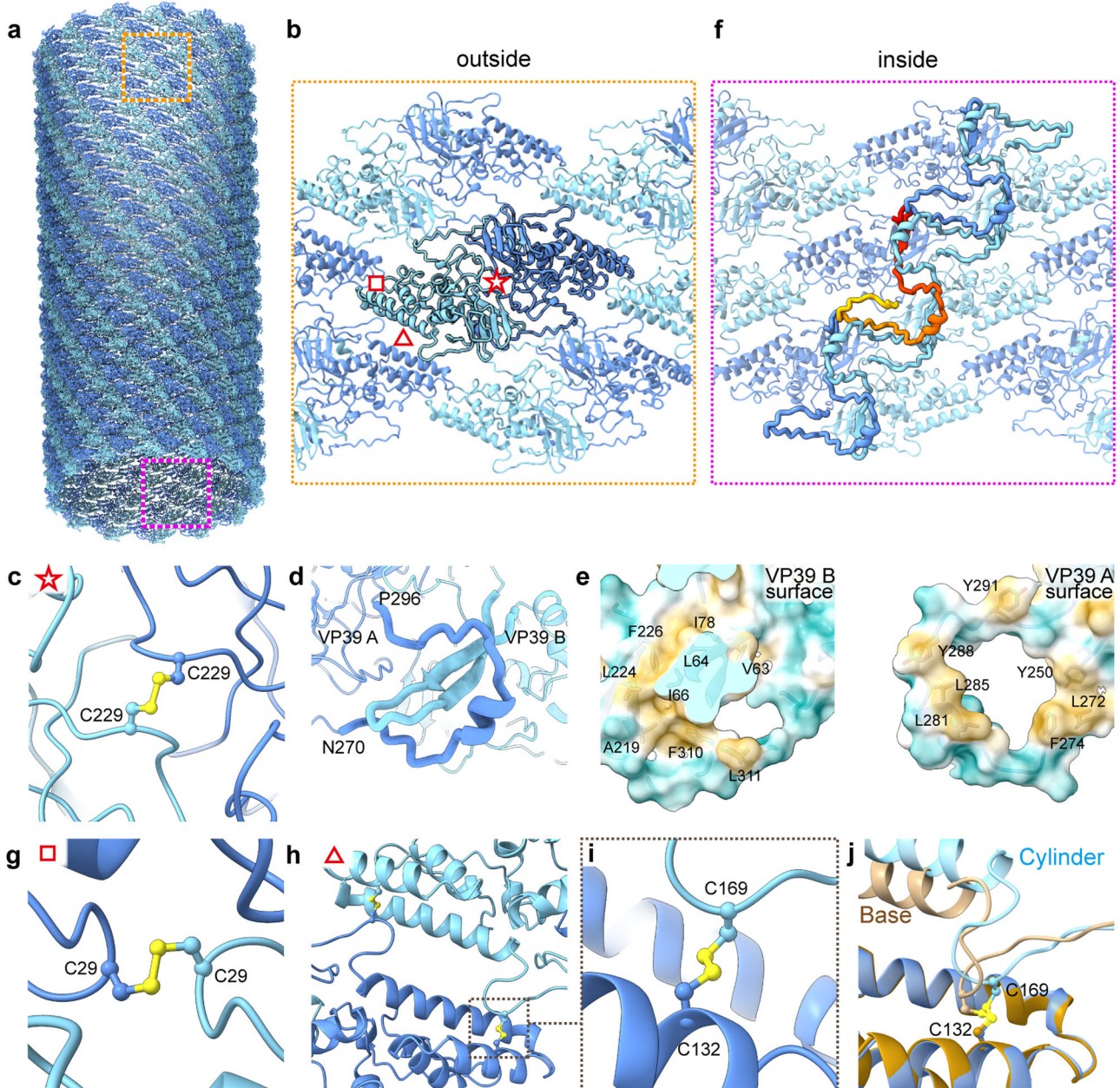

**Fig. 2 | Structural features involved in the assembly and stability of VP39 helical cylinder. a** A model for the VP39 helical cylinder. **b** An enlarged view of the orange boxed area in (**a**) shows the VP39 dimers viewed from outside. VP39A is in cornflower blue and VP39B is in cyan blue. **c** An enlarged view of the region marked by a red pentagon in (**b**) indicates the disulfide bond composed of two Cys229 from VP39A and VP39B in the same dimer. **d** The C-terminal fragment of VP39A forms a "ring"-like structure encircling the E-loop from VP39B. **e** A surface view of the interface between VP39A and VP39B, at the same location as depicted in (**d**). The left panel shows the interface surface of VP39B superimposed with the model and hydrophobic amino acids. The right panel shows the interface surface of VP39A, which is the opposite surface to that in left panel, superimposed with the model and hydrophobic amino acids. Both surfaces are rich in hydrophobic amino acids and form two corresponding hydrophobic "rings". Hydrophilic and hydrophobic surfaces are shown in cyan and goldenrod, respectively. **f** A zoomed in view of the red boxed area in (**a**) viewed from inside. The C-terminal (in yellow) binds the neighboring lower dimer, then residues in red bind the neighboring "upper" dimer, thereby forming a "S"-like structure link different dimer together. **g** An enlarged view of the region marked by a red square in (**b**) indicates the disulfide bond composed of two Cys29 from VP39A and VP39B in neighboring dimers. **h** An enlarged view of the region marked by a red triangle in (**b**) shows two VP39s from different dimers interacting with each other through two disulfide bonds composed of Cys132 of VP39A and Cys169 of VP39B. **i** A zoomed in view of the boxed area in (**h**) highlights the disulfide bond formed by Cys. **j** At the end of the helical cylinder, the disulfide bond composed of Cys132 of VP39A and Cys169 of VP39B is maintained although the distance between the dimers is reduced; VP39A and B in base are colored in dark brown and light brown respectively.

center-most portion sealing of the nucleocapsid (Fig. 1d, Supplementary Fig. 2e); further analyses revealed that the plug density has C7 symmetry.

In the apical domain, a cap rests on top of the head, protruding outwards from the capsid (Fig. 1b, Supplementary Fig. 2f). The cap has a conical shape with a height of ~150 Å and a diameter of ~ 250 Å at the bottom where it associates with the head structure (Fig. 1b and Supplementary Fig. 2f). Interestingly, a slim "column-shaped" density can be seen extending through the head into the cap (Supplementary Fig. 2f–h). In the middle of the column-shaped density, there are "wing" like densities linking it to the inner layer of head. Despite the lower resolution and not being able to determine the symmetry, the

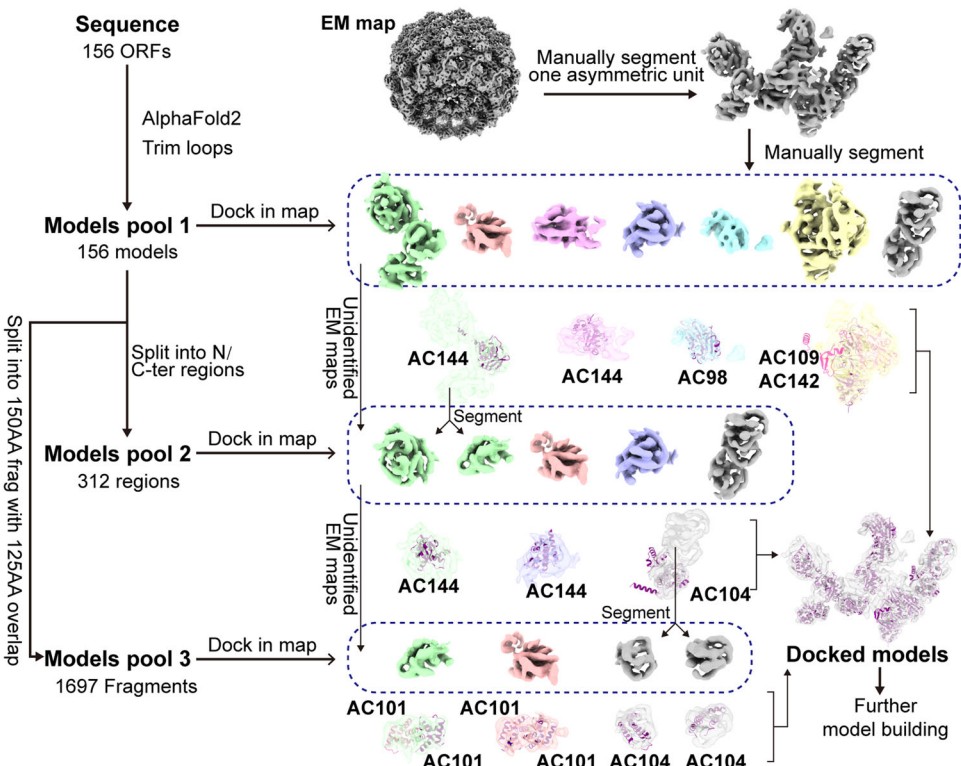

**Fig. 3 | Flowchart for modeling the ends of AcMNPV nucleocapsid.** Determining the components of the base and head proteins utilized a herein described computational modeling protocol. Models for each of the 156 ORFs were generated with AlphaFold2 and assessed for their fit to density. Progressively smaller portions of the models and density fragments were used as domains were assigned and modeled.

density is reminiscent of portal complexes found in other dsDNA-containing viruses, such as bacteriophages[35–38] and herpesviruses[39,40]. Expanding on this, we docked the T4 portal structure (PDB: 3JA7)[41] to the column-shape density in the head (Supplementary Fig. 2h). Though not an exact match to the T4 portal, and possibly having different symmetry, this density appears to have similar structural elements (crown, wing, stem, and clip) (Supplementary Fig. 2h). Based on this and its relative location in the capsid, it is likely that this density is the portal complex. However, in the absence of any additional biochemical evidence and/or high-resolution structure, the exact nature of the column-shaped density in the head remains unknown and the focus of future studies.

## Organization and stability in the AcMNPV helical cylinder

Local refinement of the main helical cylinder in the AcMNPV capsid resulted in a 3.2 Å resolution map for the major capsid protein (Supplementary Fig. 1c, and Supplementary Fig. 3a, b), which was sufficient to build a de novo atomic model for VP39 directly from the density map (Supplementary Fig. 3c). The model was then placed into the 3.9 Å helical cylinder density map to examine the overall arrangement of VP39 in the helical cylinder (Fig. 2).

The asymmetric unit of the helical cylinder is parallelogram-shaped, containing a VP39 dimer (VP39A and VP39B) with local 2-fold symmetry (Fig. 2b). Each VP39 shares a common architecture, which can be divided into three regions. The N-terminus of VP39 points toward the interior of the capsid where the N-terminal region forms a large portion of the inner surface in the helical column. This N-terminal region of VP39 (Arg14-His81) contains three loops and three short α-helixes ($\alpha_1$-$\alpha_3$). These helices are followed by one distinct two-stranded, anti-parallel β-sheet composed of strands $\beta_A$ and $\beta_B$ linked by a loop; this features is reminiscent of the extended loop (E-loop) found in the canonical HK97-fold[42,43] (Supplementary Fig. 3d). Immediately following the aforementioned "E-loop", the central region (Leu82-Asp245) contains four α-helices ($\alpha_4$-$\alpha_7$) and makes up a majority of the outer surface of the capsid cylinder. The longest helix ($\alpha_7$) contains 22 amino acids (Supplementary Fig. 3c), and again shares structural similarities to the "spine helix" in P-domain of HK97 fold[43]. However, this helix is not flanked by the long β-sheet found in the P-domain of the HK97-like fold. On the outermost portion of the cylindrical capsid, residues Phe180-Ala192 form a loop abide on a distinct four-stranded anti-parallel β-sheet. The β-sheet is composed of strands $\beta_C$, $\beta_D$, $\beta_E$, and $\beta_H$, surrounded by two short helices ($\alpha_8$ and $\alpha_9$) and additional loops. These features including the E-loop, spine helix, and four-stranded anti-parallel β-sheet, represent key structural hallmarks found in the HK-97-like fold (Supplementary Fig. 3e), and thus, we have adopted a similar naming convention for these corresponding VP39 domains. In most HK-97-like folds, the A-domain is usually orientated almost perpendicular to the E-loop, though in VP39, the A-domain lies down and resides on the E-loop. This change likely allows for VP39 to be arranged with helical symmetry and not the icosahedral symmetry typically seen in bacteriophages. The C-terminal region of VP39 (Gly246-Asn320) is primarily composed of loops and one short helix ($\alpha_{10}$) (Supplementary Fig. 3c), which wraps the E-loop of a neighboring VP39 subunit, thus contributing to the inner surface of the capsid. Additionally, residues Tyr301-Ile304 in the C-terminal region form one of the four strands ($\beta_H$) in four-stranded β-sheet in the A-domain (Supplementary Fig. 3c).

A number of interactions dominate the VP39 dimer, likely contributing to both the stability of the capsid and the establishment of the asymmetric unit. At the center of the VP39 dimer, cysteines (Cys229) from each VP39 form a disulfide bond (Fig. 2c). Another important interaction occurs at the interface between the "E-loop" of one VP39 and the C-terminal region of another VP39 (Fig. 2d). Residues Asn270-Pro296 in the C-terminal region, in addition to Tyr250, form a ring-like structure, creating a "hole" for the insertion of the "E-loop" from the adjacent VP39. Hydrophobic amino acids are located along the interface, creating and essentially tying a knot between the VP39s

in the dimer (Fig. 2e). Finally, a number of hydrogen bonds can be seen along the VP39 dimer interface (Supplementary Fig. 3f).

In addition to its roles in the assembly and stabilizing of VP39 dimer, the C-terminal region also plays important roles in the assembly of helical cylinder. Residues Arg260-Asp277 of one VP39 interact with a "lower" VP39 (the VP39 for an asymmetric unit immediately below) through a series of hydrogen bonds. Residues Asn305-Phe310 also interact with an "upper" neighbor VP39 dimer (Fig. 2f, Supplementary Fig. 3g). When taken together with the intra-asymmetric unit knot, the C-terminals of VP39 form a series of continuous interactions with an "S"-like shape, linking the neighboring helical starts together (Fig. 2f).

In addition to Cys229, which forms a disulfide bond between the two VP39s in each asymmetric unit, Cys29, Cys169 and Cys132 (Fig. 2g–i) can form disulfide bonds between VP39s from adjacent asymmetric units. These disulfide bonds lock the neighboring dimers together and are likely critical contributors in the assembly and stability of the helical capsid cylinder. In addition to the cysteines forming inter-subunit disulfides, Cys18, Cys36, and Cys49, located on the inner surface of the capsid in the density map, appear to have additional associated density and some role in interacting the dsDNA genome. A recent report suggests that Cys36 and Cys49 may form a zinc finger, supporting our observation that these VP39 cysteines play a dual role in maintaining VP39 architecture and interact with the viral genome[44].

At both ends of the AcMNPV nucleocapsid, the helical cylinder diameter is reduced by ~20% due to its affiliation with the head and base domain; interactions between VP39 and the other structural proteins appear to drive this reduction in diameter. However, the 14 helical-start patterns are upheld, and accordingly, the number of the VP39 dimers in one horizontal circle of the cylinder is maintained (Supplementary Movie 1). Fitting the model of VP39 dimers in the density at the capsid ends reveals that the disulfide bond formed by two Cys229 inside the VP39 dimer are retained. Additionally, the disulfide bond between Cys169 and Cys132 in neighboring VP39 dimers is kept despite being "compressed" (Fig. 2j). However, the disulfide bond between Cys29s, as well as some loops, are not resolvable at the ends of the capsid.

## Modeling the AcMNPV capsid ends

Distinct from the parallelogram-shaped major capsid protein VP39 dimer, additional structural features can be seen at both ends of cylinder, forming the apical and basal domains of the nucleocapsid (Fig. 1b, d). Unlike the central cylinder of the AcMNPV nucleocapsid, the resolution for the base and apical domains is insufficient to build atomic models de novo from the density, though it is sufficient to fit, assess and refine potential computational models.

The complete nucleotide sequence of AcMNPV genome contains 156 methionine-initiated open reading frames (ORFs)[9], including more than 20 potential structural proteins[45], though their precise locations remain unknown. Furthermore, there are many ORFs without annotation, which may also encode capsid-related proteins. Here, we leveraged the power of AlphaFold2[46] to predict atomic models directly from sequence to create a high throughput fitting and screening procedure when combined with the cryo-EM maps.

We generated models for all ORFs, including both annotated and hypothetical proteins, using AlphaFold2. This pool of candidate structures could then be fit to the cryoEM density map to identify and model portions of the base and head domains. For each AlphaFold2 model, low-confidence (below pLDDT of 70) residues were removed from the predicted models. The density maps of the C14 base and C7 plug (Fig. 1d) were manually segmented into a single asymmetric unit based on repeating features. This asymmetric unit was further manually segmented into seven sub-maps based on the resolvability of distinct protein "blobs" (Fig. 3). It is important to note at this point, segmentation is only based on visual assessment of density connectivity and not the total number of expected capsid proteins in the base or apical domain. Thus, these seven sub-maps may actually account for more or less than seven additional capsid proteins. Every candidate model in the pool was automatically docked into each of the seven sub-maps using the Phenix tool "*dock_in_map*". The top ten models with best map-to-model cross-correlation coefficients (CC) (Supplementary Table 1) were manually checked for the fit to density. Two AC144s (AC144-1 and AC144-3), one AC98, one AC109 and one AC142 were found to visually fit in four of the seven sub-map densities. The corresponding densities for these models were then removed from the sub-maps. Interestingly, one of the AC144 models appeared to occupy only a portion of one segmented density, leaving two other portions of the sub-map without a model. These two portions, plus the remaining three sub-maps where no suitable fit was found, left five unidentified densities for additional evaluation (Fig. 3).

No models fit the remaining five sub-maps for two possible reasons: 1) the domain and/or protein may be intrinsically flexible and therefore not easily modeled or observed in the density, or 2) initial manual segmentation may have improperly divided an intact protein into separate sub-maps. Considering this, we divided each predicted model into two half-models, roughly corresponding to an N and C terminal portion. Again, low-confidence residues for the predicted models were removed, generating another pool of candidate models (PDB pool 2 in Fig. 3). Every candidate in this pool was automatically docked into the remaining five sub-maps and checked for fit to density as before. Again, two AC144s (AC144-2 and AC144-4) and one AC104 (AC104-3) were found to fit the density for all or a portion of the sub-maps. As before, density for the fit models was removed from the sub-maps, leaving the unmodeled densities to four sub-maps (Fig. 3).

The remaining four densities had relatively small volumes when compared to the other identified regions. Interestingly though, these small domains appeared to have local C2 symmetry. In the third round of fitting of candidate models, we split each sequence into serial 150AA fragments from N to C terminal, with each fragment having a 125AA overlap with the previous fragment. As before, structures were then generated with AlphaFold2 and trimmed based on pLDDT to create third pool of candidate models (Fig. 3). Every candidate in this pool was then docked into the four remaining sub-maps and checked as before. Four AC101 domains and two AC104 domains (AC104-1 and AC104-2) fit the remaining sub-map densities.

After three rounds of screening, four "ring" like densities encircling the four AC144s remained unmodeled but were easily recognized as a part of the AC101 model. To verify the accuracy, we again docked the models from a pool of candidate 75aa models (60aa overlap with the previous fragment) to these "ring" like densities. Models for AC101 from this candidate pool localized and best fit these densities, confirming our visual interpretation. With this, all densities at the base domain and head domain (except for the cap and portal-like structure) of the capsid were identified; the individual protein models were subsequently real-space refined using the aforementioned locally refined density maps (Fig. 3, Supplementary Fig. 4).

In examining the fit of the models to the density map, there was clear agreement between the secondary structure elements of the model and density map for all subunits (Supplementary Fig. 5). Moreover, the AC109 and AC142 density maps were well resolved, and most residues were well fit in the maps. Similarly, the overall fold and bulky side chains, particularly on the long helices, were clearly resolved in the three copies of AC104, two copies of Ac101, and Ac144-1 (Supplementary Fig. 5). Among the remaining subunits, the models fit the maps well, except for some small loops in peripheral areas. To quantitatively evaluate the agreement of the models to density maps, we calculated the correlation coefficient (CC) per residue to corresponding local refined maps (Supplementary Fig. 5n) using the "validation_cryoem" function in *Phenix*[47]. Most CC of the residues were better than 0.7, though CC values of some residues in peripheral loops were worse than 0.5. Based on the agreement of the map and model, it

appears that the use of predictive models for identifying subunits in the density map is a practical and reliable method.

While the map/model agreement provides a quantitative measure for assessing subunit identification and modeling with AlphaFold, we performed additional cross-linking/mass spectrometry experiments with bis(sulfosuccinimidyl)suberate (BS³) to validate the proposed structural assignment (Supplementary Table 2). Based on the model, we predicted ~20 possible cross-links per asymmetric, assuming a linker range of 10-15 Å between the amine groups of lysines. Unfortunately, aside from verifying the inter-dimer subunit interactions along the VP39 spine helix, none of the predicted cross-links were observed in the mass spec data. Additionally, several cross-links between polyhedrins, as well as other cross-links between the proteins in the head and base, were reported in the mass spec data, but the proteins/peptides corresponding to these cross-links were either partially or completely missing in the model. As such, the cross-linking/mass spec data was unable to verify model assignment, though no evidence was provided to contradict protein assignments either.

However, our model assignment to the density map is supported by previous biochemical experiments, which identified interactions between individual structural proteins. In particular, AC98 was shown to interact with VP39, AC104[12], AC104 can for homodimers[48], AC142 interacts with AC109[20], and AC144 interacts with AC101[24]. Furthermore, mutations and/or deletions in AC98[11], AC101[13], AC104[15,48], AC109[16,17,20], AC142[13,49], and AC144[13] all resulted in defective nucleocapsid formation or localization of the nucleocapsid. These observations are all consistent with our nucleocapsid model and, when taken together with the fit of the models to the maps, provide additional validation of our method and quantitative subunit assignment (Supplementary Table 3).

## Nucleocapsid cylinder constriction

After determining the structural proteins in the base and head of AcMNPV nucleocapsid, it is possible to better understand their role in viral assembly. The base domain of AcMNPV has an overall architecture that can be divided into an outer shell, an inner layer and a plug (Fig. 4a–c, and Supplementary Fig. 2e). The head domain has an identical outer shell and a similar inner layer with the exception of AC98 (Supplementary Fig. 2f). Additionally, the head domain does not include the plug structure. The outer shell and inner layer for both the head and base are organized with C14 symmetry, while the plug is arranged with C7 symmetry (Fig. 4a, b).

Along the helical cylinder at the base (or head), three AC104 (VP80) subunits bind to the outer surface of VP39 at the end of the helical cylinder (Fig. 4c and d). Two of the three AC104s appear to form a dimer (AC104-1 and AC104-2) immediately outside of two VP39 asymmetric units. However, based on the resolvability of the density in this region, only residues Ile490-Thr664 and Asn673-Asn685 in the AC104 dimer could be seen and modeled (Supplementary Fig. 6a). Each AC104 interacts with one VP39, compacting the outer loop (Phe180-Ala192) of VP39 (Supplementary Fig. 6b and Supplementary Movie 2) to form a more "closed" interface with the four-stranded β-sheet in A-domain of VP39.

The third AC104 (AC104-3) has a similar orientation when compared to AC104-1 but is situated closer to AC109; only residues Asp463-Ile691 are resolved in AC104-3. Similar to AC104-1 and 104-2, AC104-3 interacts with the central region of VP39, though interestingly, it only interacts with VP39s in the first/final horizontal layer of the helical cylinder. While AC104-3 is largely identical to AC104-1 and AC104-2, a long helix from residues Glu467 to Phe486 runs along the outer shell and extends to AC109/AC142 (Fig. 4d). This helix contains the reported basic tract[48] and interacts with AC109 through several hydrogen bonds and salt-bridges (Fig. 4e).

AC109 sits adjacent to AC104-3 on the end of the VP39 helical cylinder and sterically blocks the elongation of the cylinder (Fig. 4a, c).

AC109 appears to alter the conformation of the Thr163-Glu176 loop, elongating the "spine helix" in P-domain of VP39s at the terminal layer of the cylinder at both ends (Supplementary Fig. 6c, and Supplementary Movie.2). This results in the formation of a disulfide bond between Cys169 of VP39 and Cys187 of AC109 (Fig. 4f).

Structurally, AC109 can be divided into three regions (Supplementary Fig. 6d). The N-terminal region (Met1-His122) is rich in β-strands and located on the outer surface of the base and head. The central region contains three helices and two β-strands; the long helix in AC104 interdigitates the space between the N-terminal and central regions of AC109 (Fig. 4d). The C-terminal region of AC109 forms a "C-shaped" structure (Supplementary Fig. 6d). At the end of C-terminal region, a helix (Ser355-Gly373) interacts with the AC101 (later defined as AC101-2), which forms part of the inner layer of the base and head, through hydrogen bonds (Fig. 4g). Additionally, the "C-shaped" structure of AC109 accommodates AC142 (Fig. 4h and Supplementary Fig. 6e) through a broad set of interactions, rich in hydrogen bonds and salt-bridges (Supplementary Fig. 6f, g).

AC142 is located at the extreme ends of the nucleocapsid (Fig. 4a–d). The N-terminal region is at the bottom of the base (or the tip of the head), while the C-terminal region of AC142 points to the helical cylinder. Between the N- and C-terminal region is a deep groove that allows the insertion of the "C-shaped" portion of AC109 (Fig. 4h), which then allows for the formation of a hetero-dimer. Additionally, AC142 extensively interacts with a neighboring AC109 through a number of hydrogen bonds and salt bridges (Supplementary Fig. 6e–g). These interactions result in fourteen AC109-AC142 heterodimers forming a "barrel-hoop" like structure, which tightens and constricts the capsid cylinder from 500 Å to ~400 Å in diameter (Supplementary Fig. 6h).

## Structure of the inner layer and plug

The inner layer of both the head and base is composed of AC101 and AC144 (Fig. 5a) arranged with C14 symmetry (Fig. 4a). AC98 (38 K) is only found in the base as no discernable density corresponding to it can be found in the head. In the base, the plug density is also composed of AC101 and AC144 (Fig. 5a) though it is arranged with C7 symmetry (Fig. 4a, and Supplementary Movie 3).

AC98 is found in the innermost portion of the base and is composed of two domains (Supplementary Fig. 7a). The C-terminal domain is structurally similar to a haloacid dehalogenase (HAD)[50,51] and faces the center of a capsid cylinder. The N-terminal domain of AC98 interacts with VP39 (Fig. 5b) and AC144 (Supplementary Fig. 7c), while also associating with genomic dsDNA (Supplementary Fig. 7b). Moreover, each N-terminus of VP39-B in the second horizonal layer of helical cylinder interacts with AC98, while the N-termini of VP39-A and AC98 interact with the bottommost layer of dsDNA (Fig. 5b, and Supplementary Fig. 7b). Based on the previous observations, the bottommost layer of dsDNA is packaged first[32], implying that AC98 and VP39 may "clamp" this portion of the dsDNA during genome packaging.

In addition to AC98, AC144 (ODV-EC27) and AC101 (BV/ODV-42) are found in the inner layer. AC101 occupies the outermost position of the inner layer and is adjacent to AC109 and AC142 from the outer shell (Fig. 4a). ~110 residues at N-terminal and ~30 residues at C-terminal of AC101 are not resolved in the map and thus not modeled. Two AC101s are found in the inner layer and are identified as AC101-1 and AC101-2 (Fig. 5a, Supplementary Fig. 7d). The N-terminal residues, composed of five helices (Glu112-Asn218) of AC101-1 and AC101-2 form a 2-fold related dimer (Supplementary Fig. 7d), and as such, we speculate that this region is the dimerization domain. Interestingly, residues Leu243-His334 from each AC101, which is comprised of four helices and connecting loops, form a ring-like structure that encircles both AC144s in the inner layer (Fig. 5c, Supplementary Movie 3). To accomplish this, the loop (Ser 219-Thr241) connecting the dimerization domain to

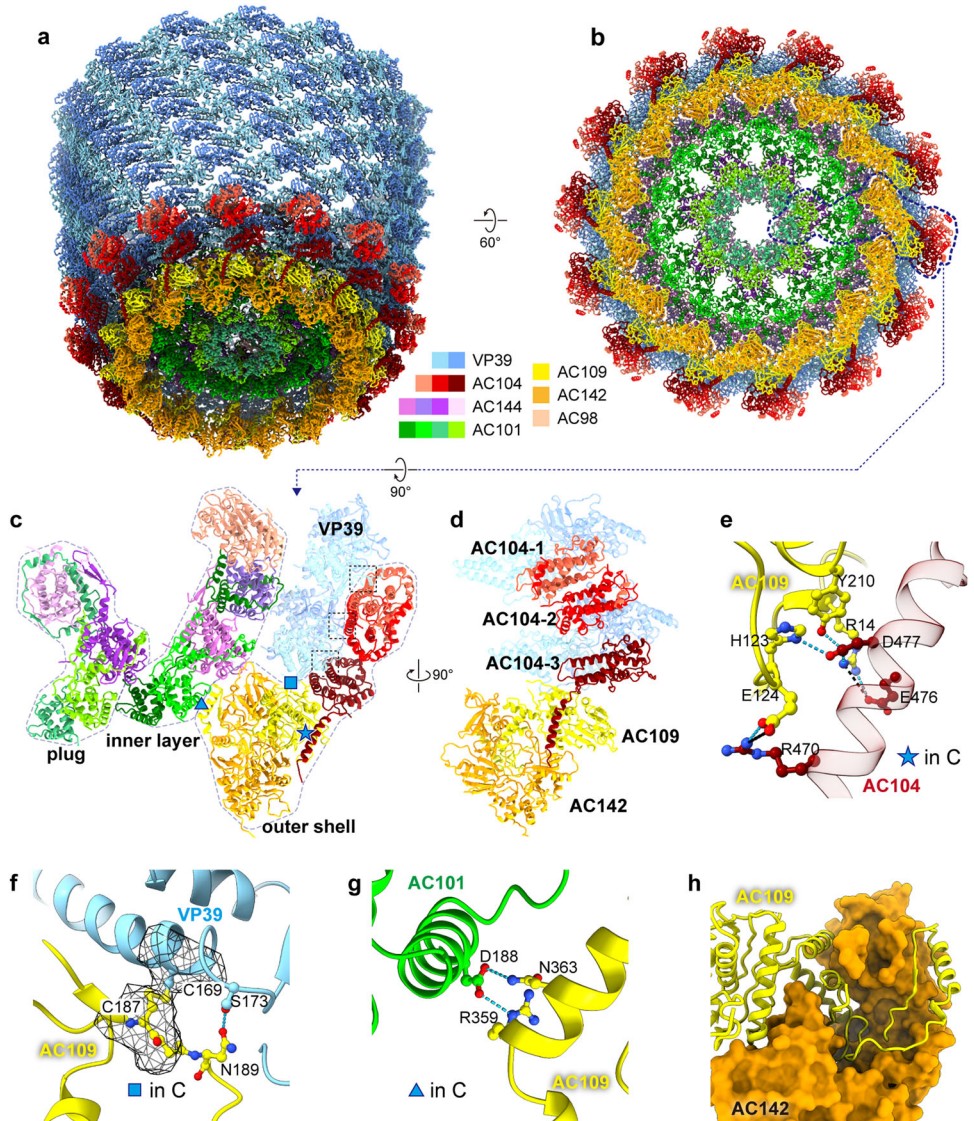

**Fig. 4 | Features of the outer shell of the head and base. a** The model for the nucleocapsid base at the end of VP39 helical cylinder is shown. **b** A top view of the base is shown. While the base and helical cylinder have C14 symmetry, the central plug is organized with seven-fold (C7) symmetry. The dotted lines indicate one asymmetric unit. **c** a model for one asymmetric unit of the base is shown. The base is composed of outer shell, inner layer and a plug. **d** The outer shell is composed of AC104, AC109 and AC142. Three AC104s are situated outside of the VP39 dimers at the end of the helical cylinder, while AC109 likely blocks the elongation of VP39 helical cylinder. **e** The area marked with "★" in panel c indicates the hydrogen

bonds (colored with cyan dotted lines) and salt bridge (black dotted line) which allows AC104 attach to AC109. **f** A zoomed in view of the area marked with "■" in panel c shows the conformation changes of VP39 in the first layer of cylinder, resulting in the formation of disulfide bond between Cys169 of VP39 and Cys187 of AC109, and a hydrogen bond with between Ser173 of VP39 and Asn189 of AC109. **g** The zoomed in view of the area marked with "▲" in panel c shows the hydrogen bonds between AC109 and AC144. **h** The deep groove of AC142 between N- and C-terminal regions allows the insertion of "C-shape" C-terminal portion of AC109. AC142 is shown in a surface representation.

the ring-like structure adopts a unique conformation in each AC101; AC101-1 interacts with the most distal AC144 (AC144-2), while AC101-2 is more compact and interacts with the proximal AC144 (AC144-1) (Fig. 5a).

Like AC101, the two copies of AC144 are nearly identical and can be roughly divided into two domains. The C-terminal domain, residues Glu62-Phe271, are encircled by AC101, while the N-terminal domain of AC144 (Cys6-Leu61) is composed of a two-stranded β-sheet and a long helix that extends away the C-terminal domain (Fig. 5d). The β-sheet in the N-terminal domain of AC144-1 extends toward the ring-like structure of AC101-1 and interacts through a series of hydrogen bonds and salt bridges to anchor the inner layer (Supplementary Fig. 7e).

Unique to the base, the plug occupies the innermost position and essentially seals one end of the capsid. Like the inner layer, each

asymmetric unit of the plug is primarily composed of two copies of AC101 (AC101-3 and AC101-4) and two copies of AC144 (AC144-3 and AC144-4), though they arranged with C7 symmetry rather than C14 symmetry found in the inner layer (Fig. 4a). The interactions among the AC101 dimers, as well as between AC101 and AC144, are almost identical to those described in the inner layer. However, the long helix (Glu27-Leu61) of AC144-3 is bent near residue Leu39, resulting in the long helix being broken into two helices: helix 1 (Ala29-Ser38) and helix 2 (Leu39-Thr60), and helix 2 is rotated ~60° compared to that of AC144-1 (Fig. 5d), allowing the N-terminal β-sheet to interact with AC101-3 and AC144-4 in the same manner seen in the inner layer, and thus anchoring AC101-3 to AC144-4 (Supplementary Fig. 7f). The rotation of helix 2 forces AC144-4 to rotate and point toward the C7 axis (Supplementary Fig. 7g), effectively sealing the base domain. Of

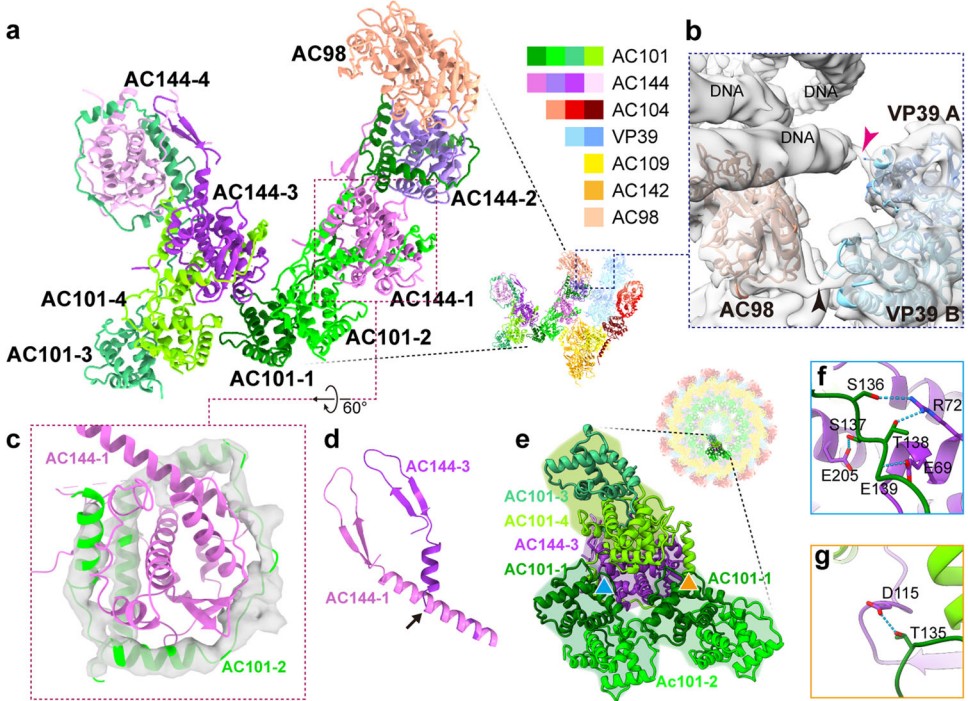

**Fig. 5 | Features of the inner layer and plug at the base of the AcMNPV nucleocapsid. a** A model of the inner layer and plug of the capsid base domain is shown. **b** Proteins AC98 and VP39 "clamp" the first dsDNA strand. The red arrow indicates the N-terminus of VP39A interacting with the DNA, while the black arrow indicates the N-terminus of VP39B in the same VP39 dimer interacting with AC98. **c** The C-terminal region of AC101 forms a "ring" like structure (green model superimposed on gray density) enclosing AC144 (pink). This structural feature occurs in both the inner layer and plug of the nucleocapsid. **d** Compared with AC144-1 (pink) in the inner layer, the long helix linking the two-stranded β-sheet of AC144-3 (purple) rotates about 60° at Leu39 (black arrow). **e** The major elements of both inner layer and plug include AC101, AC144. Here, two AC101 homodimers in inner layer clamp one asymmetric unit in the plug. **f** Seen in the area marked with a blue triangle in panel (**e**), AC101-1 from the inner layer interacts with AC144-3 of the plug through hydrogen bonds. **g** In the area marked with an orange triangle in panel (**e**), AC101-1 in the neighboring dimer interacts with the same AC144-3 with that in panel f by one hydrogen bond. The color scheme is the same as that in Fig. 4.

note, no distinguishable density could be seen before Leu39 in both AC144-2 and AC144-4.

The interactions between AC101-2 and AC109 essentially bind the inner layer to the outer shell at both ends of the capsid. Interactions between the inner layer and the plug, primarily through several hydrogen bonds, take place at the interface between AC101-1 in the inner layer and AC144-3 in plug (Fig. 5e–g). These interactions allow two neighboring AC101 dimers in the inner layer to "clamp" one AC101-AC144 complex in the plug (Fig. 5e). Hence, the AC101 dimers in the inner layer link both the outer shell and plug together, and thus stabilize the base.

## Genome organization

After symmetry relaxation, local refinement, classification and subparticle reconstructions, the densities for the dsDNA genome in both ends and cylinder could be visualized. In the base, up to seven coaxially equidistant layers of dsDNA are distinguishable (Fig. 6a)—five segments of the outermost layer of dsDNA appear to interact with the N-terminal regions of the VP39 dimers from four horizontal capsid layers (Fig. 6b). The dsDNA does not possess the helical symmetry found in the base and appears to be coaxially stacked with an interduplex distance of 27 Å (Fig. 6c, Supplementary Fig. 8a). This characteristic genome appearance is also seen in the helical cylinder (Supplementary Fig. 8b), though density at/near the head corresponding to the dsDNA is not well resolved. The organization of the dsDNA exhibits a space-efficient honeycomb topology (Fig. 6d), similar to the near-crystalline genomic packing found in many dsDNA viruses[36,40]. Consistent with previous work[32], this arrangement suggests that the genome could be first packaged coaxially near the base, with subsequent layers built along the cylinder axis.

As mentioned, ODV helical cylinders varied in length with two primary size distributions, 210 nm and 310 nm (Supplementary Fig. 2d). The diameter of the inner cavity of most helical cylinders was consistent at ~42 nm. As such, the total volume for the genomic dsDNA could be calculated to be ~29.1*10$^4$nm$^3$ and ~42.9*10$^4$nm$^3$ for the 210 nm and 310 nm nucleocapsid lengths, respectively. It has been previously reported that packaging different-sized dsDNA genomes can alter the length of the capsids[32,52]. While normal BV particles are ~330 nm in length, defective interfering particles (DIP), can undergo genetic alterations during passaging in a bioreactor resulting in an average length of ~190 nm[52–55]. However, there is no reference for the normal and defective ODV particle length. To determine if the 210 nm length particles are DIPs, we inoculated 3rd instar larvae of *Spodoptera exigua Hübner* with AcMNPV polyhedra, extracted lymph from the infected larvae and infected the sf9 cells. Once we determined the multiplicity of infection (MOI) of the first passage of BV (P1), we then used 0.001MOI of P1 BV to infect sf9 cells to rule out DIPs. After harvesting the BV(P2), we conducted the same virus proliferation and purification procedures and determined the length of the ODV particles. The 210 nm length particles were not observed, likely indicating that the 210 nm particles were DIPs.

## Discussion

Despite their relevance in modern biology and importance in agriculture, ecology, and medicine, baculovirus structure has remained unresolved. Here, we report the atomic model for the nucleocapsid of the most studied baculovirus, AcMNPV. Its architecture, from the polymorphic helical cylinder to the constriction and capping of the capsid, provides insight into the fundamental assembly and function of baculovirus capsid proteins.

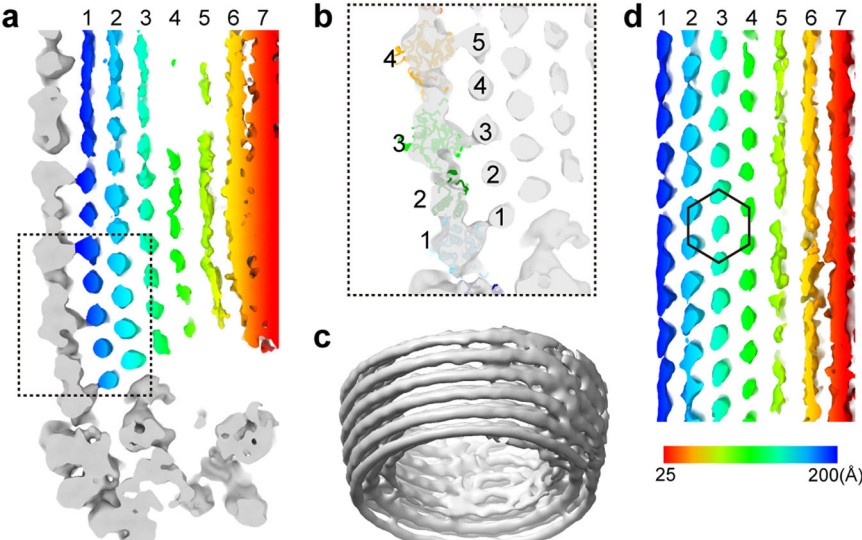

**Fig. 6 | The organization of AcMNPV's genome dsDNA. a** A longitudinal slice at the base clearly resolves seven layers of dsDNA strands. The VP39 shell and base density are shown in gray. Layers of dsDNA strands are colored individually. **b** In an enlarged view of the area boxed in (**a**), five dsDNA strands in the first layer are linked to four VP39. **c** The stack dsDNA genome (gray density) at the base suggests the dsDNA is not organized in helical symmetry. **d** A longitudinal slice of genome DNA reconstructed without any symmetry imposed indicates the crystalline honeycomb-packed dsDNA (hexagon).

## Features of VP39 give clues to evolution

Infecting a wide range of hosts including bacteria, fungi, plants, animals, and mammals, viruses with dsDNA genomes differ tremendously in their genome lengths, as well as virion size and complexity. Nevertheless, based on the structures of the major capsid proteins, dsDNA viruses can be grouped into only five classes: capsid proteins with a jelly-roll fold, double jelly-roll fold, Hk97-like fold, a four-helix bundle fold and mainly α-helical fold[56]. Structural comparison of the major capsid proteins suggests a common origins for viruses that infect hosts from different domains of life[57,58]. The major capsid protein of AcMNPV, VP39, has several common elements to that of the HK97-like fold, including the E-loop linking a two-stranded anti-parallel β-sheet, a long "spine helix" in P-domain, and a distinct four-stranded anti-parallel β-sheet in the A-domain. All of these common features hint that baculoviruses may have a common origin with the other dsDNA viruses whose major capsid proteins also have HK97-like fold, such as bacteriophages and herpesviruses. In contrast to other dsDNA viruses where the HK97-like fold is organized with icosahedral symmetry[56], VP39, with its HK97-like fold protein, is found, for the first time, to be organized with helical symmetry in baculovirus. VP39 does possess some features not previously reported in other capsid proteins with the HK97-like fold, such as the long C-terminus forming a ring structure and an A-domain that rests on the E-loop. This indicates that while VP39 may evolved from the common and robust HK97-like fold, it has introduced new elements providing for alternate pathways for the assembly, stabilization, and genome packaging in baculoviruses. As such, the structure of VP39 provides insights into how the robust HK-97 structural motif has evolved to support a wide range of needs for folding and capsid assembly.

## Modeling and cross-link mass spectrometry validation of the head and base of AcMNPV

While VP39 has previously been established as the major capsid protein, thereby making direct modeling in the density possible, no definitive protein assignments have been made for the head and base. Utilizing AlphaFold2 and density fitting presented a reasonable way to exhaustively model, evaluate and assign proteins to these portions of the nucleocapsid. The models agreed well with the density maps, showing clearly matched secondary structure and some bulky chain

chains, as well as being supported by previous biochemical work. However, direct experimental validation of these protein/density assignments could provide the best mechanism to assess our model, and as such, we performed cross-linking mass spectrometry on the AcMNPV nucleocapsids. Unfortunately, this failed to produce complete validation of our protein assignments in the head and base.

A potential issue in this experiment was the choice of the linker, BS³, which targets the amine groups in lysine side chains, in addition to having some reactivity with the hydroxyl groups in serine, threonine and tyrosine. BS³ has a spacer length of 11.4 Å and cross-linking can only occur if the atoms involved in the cross-link occur at this distance. Examining the models for potential cross-links fitting this approximate length (10-15 Å), we determined that there were less than 20 potential cross-links in an asymmetric unit. When accounting for occluded or internal cross-links, the number of exposed residue pairs dropped significantly. A single potential cross-link in adjacent VP39s was observed in the model and verified by cross-linking/mass spec data. Cross-links in AC104 between K515 from one subunit and K515, K506 and K627 from another subunit were predicted but not reported in the mass spec data. Another set of predicted cross-links between AC109 K53 and AC104 K497 and K636, as well as AC142 K141 and AC104 K469 were also not found in the mass spec data. Conversely, several cross-links in the head and base proteins were observed in the mass spec data but not predicted by our model, though all or portions of the residues involved in these cross-links were not modeled. The question arises, why were potential cross-links not observed in the mass spec data? The answer is likely that the cross-linkers were either blocked from interacting with one or both cross-linking sites or the distances required for the BS³ cross-linking were not optimal. As with all maps at this resolution, there is likely to be some small amounts of error in positioning the atoms, particularly at the ends of long side chains where the density is less well-resolved. Even 1-3 Å differences in the position of atoms (which would amount to 1-2 voxels in the density map) could alter the viability of the predicted cross-links. Furthermore, the models, particularly for AC 101 and AC104, are not complete models. As such, the addition of several hundred residues to the head and base models could potentially occlude what were interpreted as viable cross-linking sites. Additionally, to prevent breaking or distorting the nucleocapsid structure, we did not isolate the nucleocapsids

with gradient centrifugation, resulting in the presence of a considerable amount of polyhedrin proteins in the buffer. This abundance of polyhedrin has less impact on single-particle cryo-EM of the nucleocapsid, though it may affect the cross-link/mess spec experiment. In fact, we did observe multiple cross-links between polyhedrins in the mass spec data due to the large quantity of polyhedri. Ultimately, the cross-linking/mass spec experiment was simply not capable of providing more definitive validation of protein assignments. The use of different cross-linking agents may provide additional validation, though the proteins on the exterior of the virus are only partially complete and correlating cross-linking/mass spec with our models may have only limited utility. We are continuing to pursue higher resolution structures of the head and base of AcMNPV, which will help us to further elucidate their structure.

### Outer shell proteins impart capsid stability

As mentioned, the imaged AcMNPV nucleocapsid length varied. The base and head domains were essentially constant in the images of the AcMNPV, though the length of the helical cylinder between these two end structures varied. The question then arises, how is the helical cylinder terminated? Based on the AcMNPV reconstructions, it appears that the base and head domains constrict the diameter of the helical cylinder, eventually terminating the addition of VP39 layers in the cylinder. Among the structural proteins in the base and head domains of AcMNPV, AC104, AC109, and AC142 form the C14 outer shell at both ends of the capsid, with AC109 specifically blocking VP39 addition. These proteins appear to have no interaction with genomic DNA, and likely have little direct influence on genome packaging. Therefore, the proteins in the outer shell likely serve as a physical blockade in preventing cylinder elongation. Accordingly, deletion of genes encoding AC104, AC109, or AC142 resulted in single-cell infection phenotype[13,15,16,20,49].

AC104 contains an active nuclear localization signal (NLS) (residues 424 to 439)[59] and a basic tract (residues 466-488)[48]; previous studies have indicated that the conserved basic tract is essential for the assembly and stability of the capsids. Total deletion of AC104 results in failed migration of the capsid from the virogenic stroma to the nuclear periphery[59], indicating that it also plays a role in capsid transportation. Large truncations (up to 423 amino acid residues) at the N terminus of AC104 still result in virus propagation[48], though truncations of the first 499 amino acids, which include both the putative NLS and basic tract, fail to produce infectious virion[48]. In our structure, we do not see residues before Asp463, though the basic tract can be seen in AC104-3 where it passes over and interacts with AC109 and AC142. In AC104-3, it is easy to surmise that this long helix plays a critical role in linking the components of the outer shell. Given the location of the long helix of AC104-3 on the outermost surface of the base/head domains, it is conceivable that the NLS domain is also located on the outer portion of the head/base domains so as to play a role in capsid transportation after the removal of the viral envelope.

Interestingly, deletion of AC109 does not prevent budding virions production but completely abolishes infectivity; AC109 deletions do not prevent the formation of occlusion bodies, but no ODVs are packaged into the occlusion bodies[16–18,20]. As AC104 bridges AC109 and AC 142, it is conceivable that, based on our structure, AC109 deletions could still form outer shell structures, though the resulting structures would likely be abnormal, impart less capsid stability, and the transportation performed by AC104 would be seriously altered. Furthermore, the addition of an influenza HA epitope (CYPYDVPDYASL) to either the N- or C-terminus of AC109 affected function[16]. In our structure, the N and C-termini are located on the inner surface of the outer shell and interact with AC142 (Supplementary Fig. 6i–k). There is limited space to accommodate the HA tag, which would likely result in alterations at the AC109 and AC142 interface[20]. As such, the three outer shell proteins play a critical role in mediating capsid assembly and disruptions within these proteins can lead to aberrant capsid production.

### The inner layer and plug proteins point to a multifunctional role

Unexpectedly, the inner layer of both the head and base domains, as well as the plug, are primarily composed of AC144 and AC 101. The base domain contains an additional protein, AC98, located on the innermost portion of the base inner layer. Given its location on the inner layer, it would be nearly impossible for AC98 to enter the capsid interior and therefore, it is likely assembled ahead of the formation of the plug or even before the assembly of the base. Additionally, AC98 is the only protein in the base domain that has any discernible interactions with dsDNA strands (Fig. 5b). The N-terminus of AC98 appears to contact the first detectable dsDNA strand (Supplementary Fig. 7b), suggesting that AC98 might be involved in the packaging of genomic DNA. Supporting this, numerous empty tubular structures without genomic DNA have been previously detected within the virogenic stroma where AC98 was deleted from the viral genome[11]. However, the N-terminal domain of AC98 also contains a putative HAD domain and deletion of residues at the active center of HAD domain blocked the dephosphorylation of P6.9, a key step in genome packaging, resulting in empty tubular structures[51]. Unfortunately, P6.9 could not be identified in our density maps. While the HAD superfamily does contain ATPases, it also includes, phosphoesterases, phosphonatases, and other enzymatic activities[50]. Thus, it is unclear if AC98 is indeed the ATPase that powers genome packaging or just a key component in genome packaging. Previous experiments also indicated that AC98 interacts with AC104[12], though no obvious interactions were observed in our structure. Given that we are only resolving a portion of the AC104 structure and its relative proximity to AC98, as well as only reconstructing complete virions, it is conceivable that there are direct interactions between the two proteins, indirect interactions through their association with VP39, or interactions during virion assembly.

Previous studies have shown that no well-defined capsid could be identified when AC144 and AC101 were deleted[13]. However, both AC101 and AC144 have been implicated in having additional functional roles. AC144 was initially detected on both the ODV envelop (E) and capsid (C) with a molecular mass of 27 kDa, and consequently was named ODV-EC27[60]. Amino acids 1-110 of AC144 demonstrate ~25%-30% sequence similarity with cellular cyclins within the cyclin box, as well as an association with either cds6 or cdk6 in vitro. The cdk6-AC144 complex is associated with the proliferating cell nuclear antigen (PCNA)[61]. A putative NLS sequence, $^{357}$KRKK$^{360}$, was also found in AC101 and its deletion abolished actin polymerization activity in the nuclei[25], a critical step in baculovirus transport. However, the region containing the NLS in AC101 is not resolved in the map and therefore not modeled. AC101 has also been shown to interact with AC144 and P78/83[24]; however, no resolved density could be attributed to P78/83; this lack of resolved density could be due to a number of reasons including that P78/83 may have been lost during the purification process. Together, this points to the fact that the multiple copies of the AC101 encircling AC144 in the inner layer and plug likely play a multifunctional role in the viral lifecycle.

### AcMNPV apical domain is involved in genome packaging

Unlike genome packaging in DNA viruses with linear genomes which have been relatively well characterized, packaging of circular dsDNA is less well-understood. The widely accepted model for the circular dsDNA genome packaging in AcMNPV was proposed by Dr. M.J. Farse in 1986[32]. In this model, the circular genome is wound and packaged into the preassembled "capsid sheath" through the cap. The viral genome is driven and packaged by an ATP-driven molecular motor portal, as seen in bacteriophages and herpesviruses[62,63]. While a concrete identification of a portal complex in AcMNPV could not be established, the fact that the portal-like structure has similar structural

elements to portal complexes from bacteriophages[41,64] and herpesviruses[40] provides some evidence to support the existence of a portal complex in AcMNPV. Additionally, most bacteriophages or herpesviruses have additional structures on the outer portion of the portal complex, such as a cap in Herpes Simplex Virus type-1 (HSV-1)[40,64] that may anchor the genome terminus. In AcMNPV, a cap sits on the top of the capsid; these caps were easily lost during purification, and in micrographs genomic dsDNA could be seen emptying from the capsids. This indicates that the cap of AcMNPV capsid may perform a similar function to that of portal cap from HSV-1. While we cannot resolve the structure of the portal and cap to a sufficient resolution for modeling, a number of studies have indicated that the AC09(P78/83)[24], AC53[65], AC10(PK1)[66], AC77(vlf-1)[67], AC102[68], and P6.9[69] are structural proteins involved in some aspect of baculovirus assembly and genome package. We can speculate that some of these proteins may indeed belong to the portal-like complex or cap, though we simply lack the ability to clearly assign these putative structural proteins in our map currently. Further studies are needed to resolve their structure and role in capsid assembly and genome package.

### Identifying the components of the head and base

While clear, high-resolution density in the helical cylinder made it possible to build atomic models directly from the density map, our lack of overall resolvability and lack of protein assignments at both ends of AcMNPV nucleocapsid precluded us from building de novo atomic models. Typically for generating accurate structural models from near-atomic resolution density maps, clearly interpretable density and a corresponding sequence for each protein in the complex is required. Assuming the resolution is sufficient to clearly define the path of the protein, generation of the model can be made directly from the density map using a variety of bottom-up[70,71] or top-down[72] approaches. However, we may neither know the exact components – and therefore the exact protein sequence – present in the complex, nor have the resolution to provide an unambiguous trace of the interested proteins in the complex. However, the recent advancements in machine learning and predictive modeling have made it possible to reliably build and assess models in intermediate-resolution structures.

In this work, we leveraged the power of machine learning in predicting the structure of all possible ORFs in the baculovirus genome. While many of these predicted models are low-confidence models, enough of the resulting models produced a pool of structures that could be used to assess and assign protein sequence/structure to previously unmodeled portions of the density map. This requires that 1) there is a set of possible models that can be generated with reasonable levels of confidence from a finite set of possible sequences, and 2) while the density does not have to clearly resolve all structural features, there is sufficient resolvability in the map to discriminate model fit density. As such, this approach may not be universally applicable, but does provide a potentially useful application of AlphaFold2 for model building in intermediate-resolution density maps from cryo-EM.

## Methods

### Virus propagation and capsid purification

AcMNPV was propagated by rearing the third instar larva of *Spodoptera exigua* with an artificial larval diet containing budded virions (BVs). About five days later, the dead larvae infected with AcMNPV were collected and homogenized. After several rounds of centrifugation and homogenation, occlusion bodies (OBs) were purified and kept in 0.1 M Sodium Phosphate Buffer (PBS). OBs were then treated at 37°C for 3 minutes with a buffer containing 0.3 M $Na_2CO_3$, 0.5 M NaCl and 30 mM EDTA at pH11. The alkaline lysis was stopped by adding about 3× of PBS buffer, and then centrifuged at 2000g for 10 minutes to discard the debris of the OBs. The supernatant was then centrifuged at 20,000 g at 4 °C for 30 min. The precipitation was resuspended with PBS and kept at 4 °C for further use.

### Preparation of cryo-EM samples and data collection

Cryo-EM grids were prepared as previously described[73]. Briefly, 3 μL of samples were applied to newly glow discharged R1.2/1.3 holey copper grids (Quantifoil Micro Tools GmbH, Jena, Germany). The grids then were blotted, and flash-frozen in precooled liquid ethane using a Vitrobot Mark IV machine (Thermo Fisher Scientific, Waltham, USA) at 100% humidity. Cryo-EM data was collected on a Titan Krios 300 kV electron microscope (Thermo Fisher Scientific) equipped with a K3 camera (Gatan) in super-resolution mode. Automatic data collection was performed using SerialEM[74] with defocus values ranging from 1.0 μm to 3.0 μm. Movies were recorded with 48 frames with a total dose of ~60 e⁻/Å². The nominal magnification was 64,000× giving a calibrated pixel size of 1.35 Å. Two datasets were collected on two microscopes with the same acquisition conditions.

### Cryo-EM data processing

Beam-induced motion correction and dose weighting were performed by MotionCor2[75]. Contrast transfer function parameters were estimated by CTFFIND4[76]. Virion helical cylinders were automatically picked using the SPHIRE-crYOLO filament mode[77] and then divided into overlapping segments. Several rounds of reference-free 2-D classification using Relion3[78] were used to remove ice contamination and deformed virus particles. The remaining particles were refined and reconstructed using cryoSPARC[79] with helical symmetry imposed. The initial reconstruction was further processed using 3-D classification without image alignment in Relion3 in order to differentiate heterogeneous radius. The class with the highest resolution and clearest features was selected and further refined with cryoSPARC, generating a helical cylinder density map with a resolution of 3.9 Å. The local defocus and astigmatism for the symmetry-relaxed data were further refined; rotation and shift parameter were also locally refined using cryoSPARC. The resolution after the local refinements was determined to be 3.2 Å using the gold-standard FSC at the 0.143[80,81].

Both ends of the virion were manually picked from the micrographs simultaneously, followed by several rounds of reference-free 2D classification in Relion3. Particles were refined against an initial model with cryoSPARC. 3D classification without image alignment was then performed by Relion3, yielding two different EM-maps, one for head and the other for the base. Both maps showed C14 features and were further refined by imposing C14 symmetry with cryoSPARC heterogeneous refinement, resulting in a map at a resolution of 7.4 Å for head and a map for the base at 7.0 Å resolution. Smeared densities, likely caused by imposing incorrect symmetry, were observed in the center of both the base and head. The end particles were then symmetry expanded and processed for 3-D classification with a mask for the center part of the head and base using Relion3. The base domain was shown to have C7 symmetry. However, we could not resolve the symmetry of the central part of head. The head and base domains, outside of the central density features appeared to be nearly identical. Therefore, the head and base data were combined and locally refined; the reconstruction was determined to be ~4.3 Å resolution. The base and the plug were then locally refined, and the reconstructions were determined to be 4.8 Å and 4.9 Å resolution, respectively. These three local refinements were performed with cryoSPARC.

In order to resolve the organization of the genome dsDNA, the VP39 signal was subtracted from helical cylinder particles. The subtracted particles were then symmetry expanded. 3D classification with column-shaped mask, whose radius corresponded to the DNA layers, was performed using Relion3, resulting in a density map of the dsDNA genome inside the capsid. The base and head particles were symmetry expanded and 3D classified using the same methods, generating a

density map of the DNA inside the base. The genomic DNA in the head was not clearly resolved.

The Fourier Shell Correlation (FSC) curves (Supplementary Fig. 1c) were calculated using a tight mask alone and a tight mask with correction by noise substitution, respectively, with two half-maps in cryoSPARC. The resolution estimations for all cryo-EM density maps were calculated according to gold-standard FSC at the 0.143[80,81]. The local resolutions were determined using cryoSPARC.

The unwrapped maps were generated using EMAN2[82]. All figures and movies were prepared in UCSF *ChimeraX*[83] and *Chimera*[84].

## Model building

The VP39 model was built de novo in *Coot*[85] and refined with *Phenix*[47]. The AC98, AC101, AC104, AC109, AC142 and AC144 models were initially built using AlphaFold2 and fit into the corresponding density with the "process_predicted_model" and "dock_in_map" modules in *Phenix* (Fig. 3). Models were then manually adjusted according to the corresponding density map in Coot and refined with *Phenix* and *Rosetta*[86]. The statistics of model refinement and validation are listed in Supplementary Table 4.

## Cross-linking and mass spectrometry analysis

The purified AcMNPV nucleocapsid was cross-linked by reacting it with bis[sulfosuccinimidyl] suberate (BS$^3$) (Thermo Fisher Scientific) at different concentrations on ice for 2 h and terminated by using 50 mM Tris-HCl pH 7.5 at room temperature for 15 min. Then the samples were further checked through SDS-PAGE to find the optimal cross-linking condition. The final concentration of the cross-linker was 2 mM. After terminating the reaction, the sample was ionized and directly introduced into a Q-Exactive mass spectrometer using a nano-spray source. Survey full-scan MS spectra (from $m/z$ 300 to 1800) were acquired using an Orbitrap analyzer with a resolution $r = 70,000$ at an $m/z$ of 400. Cross-linked peptides were identified and evaluated using the pLink2 software[87].

## Volume calculation for the capsid internal cavity

To get a precise length of the capsid internal cavity, we shifted the start-end coordinate determined by SPHIRE-crYOLO towards both sides. Then, we extracted the particles and aligned them by 2D classification. The start/end point coordinates of the internal cavity were measured in 2-D class average and converted to their coordinates in micrographs. Discarded the bent and incomplete particles in the micrographs, we finally got the length values for 2503 intact particles (Supplementary Fig. 8c). The length values were then categorized with a step size of 5 nm and were displayed in the length distribution histogram (Supplementary Fig. 2d). The internal volume of AcMNPV was calculated by cylinder volume formula ($\pi r^2 h$), with internal $r = 210$ Å (Supplementary Fig. 2c). The genome size (kb) to volume ($10^4$ nm$^3$) ratio is around 4.87, estimated based on previous reported internal volumes of bacteriophage capsids[38] by linear regression.

## Reporting summary

Further information on research design is available in the Nature Portfolio Reporting Summary linked to this article.

## Data availability

The cryo-EM maps of the overall helical cylinder, apical domain, and base, have been deposited in the Electron Microscopy Data Bank (EMDB) under accession codes EMD-35242, EMD-35243, and EMD-35244, respectively. The atomic coordinates were deposited in the RCSB Protein Data Bank (PDB) under the accession codes 8I8A (VP39), 8I8B (outer shell and inner layer) and 8I8C (plug), with their maps in EMDB under accession code EMD-35245, EMD-35246, and EMD-35247, respectively. Previous reported atomic models of T4 phage and HK97 can be accessed under PDBID accession codes 3JA7 and 3E8K. The

length distribution raw data presented in Supplementary Fig. 2d in this study is provided in the Source Data file. The data that support this study are also available from the corresponding authors upon request. Source data are provided with this paper.

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

## Acknowledgements

The authors would like to thank the members of EM facility lab at State Key Laboratory of Biocontrol, the School of Life Sciences, Sun Yat-sen Universit; and SUSTech cryo-EM facility center, Southern University of Science and Technology, China; the staff members of the Mass Spectrometry System at the National Facility for Protein Science in Shanghai (NFPS), Zhangjiang Lab, China, for providing technical support and assistance in data collection and analysis. We are much obliged to Prof. Edward H., Egelman from University of Virginia and Prof. Rui Zhang from Washington University, for valuable help on helical reconstruction; Prof. Meijin Yuan of School of Life Sciences, Sun Yat-sen University for valuable discussions regarding the manuscript. This work was supported by the Science and Technology Planning Project of Guangdong Province, China (No: 2021B1212040017) to Q.Z.

## Author contributions

X.J., Y.G., and Q.Z. developed the original hypothesis and designed the experiments. Y.G. and X.J. undertook the cryo-EM data collection and processing, reconstruction, map refinement and atomic modeling. Y.H., L.S., S.L., X.Z., and Q.Z. cultured and purified virus. X.J., Y.G., H.V., M.L.B., and Q.Z. performed data analysis. K.Y. and W.W. provided biological insight and analysis on baculoviruses. H.L., Y.L., and J.H. performed the initial screening and cryo-EM data collection on Talos F200C. X.J., Y.G., and Q.Z. wrote the initial manuscript. Q.Z. and M.L.B supervised the project and revised the manuscript.

## Competing interests

The authors declare no competing interests.
