## [Peer Review File · Nature Communications]

Architecture of the baculovirus nucleocapsid revealed by cryo-EMREVIEWER COMMENTS

Reviewer #1 (Remarks to the Author):

Summary

In this report the authors determine the structure of the nucleocapsid of the baculovirus *Autographa californica* multiple nucleopolyhedrovirus (AcMNPV). To determine the structure the authors purified virions from occlusion bodies (OBs) isolated from cadavers of infected *Spodoptera exigua* larvae. The virions were alkali liberated from OBs, pelleted and resuspended in PBS. Virion suspensions were placed on cryo-EM grids and a cryo-EM was used to image for data collection. Data was processed and models were developed. Beyond the major capsid protein of VP39, AlphaFold2 was used to develop models for all predicted AcMNPV proteins. These models were used to map to the observed densities from the cryoEM and virion proteins were mapped to the nucleocapsid model and protein interactions were constructed for the Cap, body and tail of the nucleocapsid. Various predictions from this model were made for the cap and tail assembly, such as inhibiting VP39 assembly, and constricting the ends of the nucleocapsid cylinder. Measurements of the DNA packing within the nucleocapsid structure were determined as well and it was shown to follow similar packing as found in herpesvirus and bacteriophage.

Comments

AcMNPV is the most well studied baculovirus and has been used as a biopesticide, as well as been extensively utilized in the biotechnology industry for protein expression and is also being used experimentally for gene therapy experiments. However, very little is known about the structure of any baculovirus proteins with the exception of GP64, POLYHEDRIN, P33 and ODV-E56. This study therefore is a significant step forward and will provide an excellent base for further understanding of baculovirus development, including the key steps of nucleocapsid formation, nucleocapsid nuclear egress and entry, as well as packaging of nucleocapsids into OBs within the nucleus. The manuscript is clearly written and follows a logical flow and the comments are supported by the data and extended data presented. This study will be of significant interest to laboratories working with baculoviruses or DNA viruses in general.

Line 7. Replace "largely a mystery" with unsolved, or something similar.

Line 9. Replace "Virus" with nucleocapsid.

Line 14 and throughout the text. By convention, protein names are upper case and gene names are lower case italics. Ac104, Ac142 etc. should therefore be AC102, AC142 etc. Such as the authors use for “VP39” line 19.

Line 29. There are not several hundred species. The authors should consult the latest ICTV statement for the most recent exact number which is about 97.

Line 38. Delete “species of the”.

Line 50. Actual name for C42 is BV/ODV-C42, C42 is only an abbreviation. See Braunagel et al. J Virol. 2001, 12331-12338.

Line 51. “envelope protein”. Gp64 should be GP64, polyhedra should be polyhedrin.

Line 120. What does “an obviously unique” mean?

Line 213-218 and discussion. AC144 is also known as ODV-EC27 which should be noted. In addition ODV-EC27 has been reported to be both an occlusion derived virus (ODV) nucleocapsid and envelope protein. Do the models predicted from the cryoEM and AlphaFold1 fit with this data? This needs to be discussed.

Line 361-364. The differential nucleocapsid size could be due to presence of Defective interfering particles. Their significant presence in this study could be due to how the laboratory stocks of viruses were propagated. This needs to be discussed or ruled out.

Line 368. See comment above Line 7.

Line 458. The assembly of “a” viral nucleocapsid....

Reviewer #2 (Remarks to the Author):

In, Novel architecture of the baculovirus nucleocapsid revealed by cryo-EM, the authors describe their work determining the structure of the *Autographa californica* multiple nucleopolyhedrovirus baculovirus using single particle cryo-EM. The structure has a number of symmetry mismatches, and impressively, they were able to split the structure into multiple subparticles that they reconstructed individually. Using this approach, they were able to tease apart the individual proteins that assemble to make the overall structure. One of the most impressive aspects of the manuscript is the authors' approach of combining AlphaFold2 model building with segmentation and identification of unknown lower-resolution parts of their maps. They manage to make some pretty convincing atomic models of regions that would otherwise be uninterpretable. At the same time, this is the biggest shortcoming of the paper. It is likely that their models are correct, but they need some extra validation especially given the low resolution of the map and the fact that the densities could not be unambiguously identified. Perhaps some cross-linking mass spectrometry to validate their models? To be clear, I am enthusiastic about their approach, but there needs to be some additional validation.

Other comments in no particular order

1) The movies are a very nice addition, and really help with understanding the structure.

2) Page 6: The figure for the putative portal needs to be improved. Also, the discussion of the putative portal is somewhat ambiguous. Some of that may be by design, but that section of the manuscript seems somewhat tacked on.

3) Fig 1: What is the structure of the region between b and c and c and d?

4) P13 line 340 -refine > refinement

Reviewer #3 (Remarks to the Author):

The manuscript "Novel architecture of the baculovirus nucleocapsid revealed by cryo-EM" describes the first high resolution structure of a baculovirus capsid. The authors used a combination of helical reconstruction, single particle analysis, symmetry relaxation, and focused reconstruction to determine structures of the central cylindrical region of the capsid and the two ends. When density quality was

good enough, the authors were able to build atomic models directly into their maps. To complete the model and identify additional structural proteins, the authors used alphafold2 to fold various ORFs, and then used their experimental maps as filters to select the correct model. The manuscript is clearly written, with beautiful figures and movies. Some of most interesting findings were unexpected similarities to structural proteins in tailed dsDNA bacteriophages. In particular, the major capsid protein resembles the HK97-fold that all dsDNA phages use as their capsid proteins, and they report a protein at one end that looks a phage portal protein. In addition to the evolutionary relationships such similarities suggest, the overall architecture of the capsid is somewhat unique, providing new insights into the geometry of virus capsids.

Given the unexpected similarities to phage proteins, the paper could improve by elaborating on the suggested evolutionary relationships. For example it is difficult to tell how much the capsid protein resembles the HK97 fold. I understand the structural homology is limited, but is it possible to show a superposition of homologous regions/domains? Do the proteins share the same topologies, which might suggest divergent evolution, or do they only position certain secondary structural elements in similar ways (e.g. convergent evolution). Additionally, it is difficult to tell how well the T4 portal fits in their density, and it is confusing that they dropped a 12-fold symmetric structure in 7-fold symmetric density. Again, some sort of super-position, or topology diagram would be helpful. The authors might also want to take a look at a recent PNAS paper about archaeal spindle-shaped viruses; the authors of that paper found a protein that seems quite similar to phage portals, and that protein is a heptamer. Similarly, the authors sensibly suggest that these viruses may package DNA like phages, i.e. build an empty shell and then pump the DNA into that shell. The ATPases that power packaging are usually easy to identify via bioinformatics. Is there an obvious candidate for the packaging ATPase? Finally, do the authors have any thoughts about a circular dsDNA would be packaged using this strategy? Is the DNA linearized for packaging? If not, is the channel for the portal large enough to allow passage of a circular dsDNA.

Reviewer #1 (Remarks to the Author):**Summary**

In this report the authors determine the structure of the nucleocapsid of the baculovirus *Autographa californica* multiple nucleopolyhedrovirus (AcMNPV). To determine the structure the authors purified virions from occlusion bodies (OBs) isolated from cadavers of infected *Spodoptera exigua* larvae. The virions were alkali liberated from OBs, pelleted and resuspended in PBS. Virion suspensions were placed on cryo-EM grids and a cryo-EM was used to image for data collection. Data was processed and models were developed. Beyond the major capsid protein of VP39, AlphaFold2 was used to develop models for all predicted AcMNPV proteins. These models were used to map to the observed densities from the cryoEM and virion proteins were mapped to the nucleocapsid model and protein interactions were constructed for the Cap, body and tail of the nucleocapsid. Various predictions from this model were made for the cap and tail assembly, such as inhibiting VP39 assembly, and constricting the ends of the nucleocapsid cylinder. Measurements of the DNA packing within the nucleocapsid structure were determined as well and it was shown to follow similar packing as found in herpesvirus and bacteriophage.

Comments

AcMNPV is the most well studied baculovirus and has been used as a biopesticide, as well as been extensively utilized in the biotechnology industry for protein expression and is also being used experimentally for gene therapy experiments. However, very little is known about the structure of any baculovirus proteins with the exception of GP64, POLYHEDRIN, P33 and ODV-E56. This study therefore is a significant step forward and will provide an excellent base for further understanding of baculovirus development, including the key steps of nucleocapsid formation, nucleocapsid nuclear egress and entry, as well as packaging of nucleocapsids into OBs within the nucleus. The manuscript is clearly written and follows a logical flow and the comments are supported by the data and extended data presented. This study will be of significant interest to laboratories working with baculoviruses or DNA viruses in general.

A: Thanks for the positive comments.

Line 7. Replace "largely a mystery" with unsolved, or something similar.

A: Thank you for the suggestion, we have modified it in the revised version.

Line 9. Replace "Virus with nucleocapsid.

A: Thank you for the suggestion; we have replaced "virus" with nucleocapsid in the revised version.

Line 14 and throughout the text. By convention, protein names are upper case and gene names are lower case italics. Ac104, Ac142 etc. should therefore be AC102, AC142 etc. Such as the authors use for "VP39" line 19.

A: Thank you for the suggestion; we have changed "Ac" to "AC" for all protein names in AcMNPV in the revised manuscript.

Line 29. There are not several hundred species. The authors should consult the latest ICTV statement for the most recent exact number which is about 97.

A: Thank you for the suggestion. We have checked the latest ICTV statements and found that 103 species are currently listed in the statements. We have changed "several hundred species" to "about 100 species".

Line 38. Delete "species of the" .

A: Thank you for the suggestion. We have deleted this in the revised version.

Line 50. Actual name for C42 is BV/ODV-C42, C42 is only an abbreviation. See Braunagel et al. J Virol. 2001, 12331-12338.

A: Thank you for the suggestion. We have changed "C42" to be "BV/ODV-42" in the revised version.

Line 51. "envelope protein" . Gp64 should be GP64, polyhedra should be polyhedrin.

A: We have changed these names according to the suggestions in the revised version

Line 120. What does "an obviously unique" mean?

A: We have modified the description in the revised version.

Line 213-218 and discussion. AC144 is also known as ODV-EC27 which should be noted. In addition ODV-EC27 has been reported to be both an occlusion derived virus (ODV) nucleocapsid and envelope protein. Do the models predicted from the cryoEM and AlphaFold1 fit with this data? This needs to be discussed.

A: Thank you for the reminder. We have indicated that AC144 is known as ODV-EC27 in the abstract, as well as on lines 66 and 332 in the manuscript. We also added more discussion on AC144 in lines 479-481. The AlphaFold2 models and the cryoEM there is a long helix in AC144 (Glu27-Leu61) (see line 344-361); it is possible that this helix could be a transmembrane helix and anchor the envelope. However, we have not addressed this point as we haven't resolved the envelope structure of ODV.

Line 361-364. The differential nucleocapsid size could be due to presence of Defective interfering particles. Their significant presence in this study could be due to how the laboratory stocks of viruses were propagated. This needs to be discussed or ruled out.

A: In order to test whether the short particles are defective interfering particles, we re-conducted the infection experiment. We inoculated 3rd instar larvae of *Spodoptera exigua Hiibner* with AcMPV polyhedra from the same batch used in the manuscript. After 5 days post inoculation, we extracted several microliters of lymph and infected sf9 cells. About 5 days post infection, we took the sf9 culture media, which should contain the first passage's BV (P1) and determined the MOI. Then, we used 0.001 MOI of P1 BV to infect the sf9 cells and harvested the culture media, which contained the second passage's BV (P2). As the MOI was very low (0.001), the defective interfering particles should not be replicate without the help of normal particles. Then, we used the P2 BV to inoculate 3rd instar larvae of *Spodoptera exigua Hiibner*. ~6 days post inoculation, we harvested the dead larvae, purified the polyhedra and ODV and re-determined the length of the ODV particles using the same methods described in the manuscript. The result revealed the length of the particles were contained in a single distribution; the lengths of most particles were a little bit shorter than 310nm. We counted ~400 ODV particles and the distribution peak was centered around 280nm. (In order to avoid confusion, we didn't describe the length calculation this time.). The present results reveal the length of the ODV particles can be varied within a specific limit, and the shorter particles reported in the manuscript are most likely the defective interfering particles.

We greatly thank the reviewer for the reminders and professional suggestions. We have revised the manuscript.

Line 368. See comment above Line 7.

A: Thank you for the suggestion; we have modified it in the revised version.

Line 458. The assembly of "a" viral nucleocapsid....

A: Thank you for the suggestion, we have modified it in line 469 of the revised version.

Reviewer #2 (Remarks to the Author):

In, Novel architecture of the baculovirus nucleocapsid revealed by cryo-EM, the authors describe their work determining the structure of the *Autographa californica* multiple nucleopolyhedrovirus baculovirus using single particle cryo-EM. The structure has a number of symmetry mismatches, and impressively, they were able

to split the structure into multiple subparticles that they reconstructed individually. Using this approach, they were able to tease apart the individual proteins that assemble to make the overall structure. One of the most impressive aspects of the manuscript is the authors' approach of combining AlphaFold2 model building with segmentation and identification of unknown lower-resolution parts of their maps. They manage to make some pretty convincing atomic models of regions that would otherwise be uninterpretable. At the same time, this is the biggest shortcoming of the paper. It is likely that their models are correct, but they need some extra validation especially given the low resolution of the map and the fact that the densities could not be unambiguously identified. Perhaps some cross-linking mass spectrometry to validate their models? To be clear, I am enthusiastic about their approach, but there needs to be some additional validation.

A: Thank you for the comments and suggestions. We have revised the manuscript to better emphasize and summarize the validation of our map/model assignments.

Since the original submission of the manuscript, we have continued to optimize and process the data, as well as add additional data, resulting in improvements in the local resolution for a number of the protein subunits. In the newly added Supplementary Figure 5, the models are shown superposed on the corresponding portions of the density map. Improvement in resolution and resolvability have allowed us to clearly visualize secondary structure elements, connectivity within the protein and even large sidechain density. As seen in the figures, these structural elements match exceedingly well with the refined models and provide improved model validation. It should also be noted that the procedure we implemented when fitting the models to the maps compares all predicted models to all unassigned density. What we report is essentially the optimal model fit to the density based on an exhaustive comparison, providing a quantitative measure of map/model agreement. While not biochemical validation, this exhaustive fitting coupled with the agreement of local structural features provides method and model validation.

As suggested by the reviewer, we did perform cross-linking mass spectrometry to validate our computational modeling. Unfortunately, the results proved only moderately useful. Cross linking in VP39 was identified and corresponded to peptides along the central portion of the spine helix and the loop at the end of the spine helix. Across dimers, the spine helix loop extends over the spine helix of a neighboring dimer, likely resulting in inter-VP39 dimer interactions that stabilize the nucleocapsid. Cross-links were also identified in the AC104 and AC144 subunits, however all or portions of these peptides were not present in the predicted models. We are continuing to explore additional cross-linking approaches, as there are a number of other structural proteins potentially in the nucleocapsid that have yet to be assigned and modeled.

Despite the limited results from the cross-linking, we searched the literature to see if we could find any additional supporting information. A short summary is listed below:

1. AC98 has been shown to interact with VP39, AC104¹; Interactions between AC98 and VP39/AC144 are observed in our structure. The predicted model for AC104 represents only a small fraction of the total protein sequence. Based on the approximate location of AC104 and AC 98, interactions could be possible. Deletions of AC98 result in tub-like VP39 structures, supporting its assignment to the ends of the nucleocapsid.
2. AC101 was associated with AC144 and pp78/83² which is shown to localize to the basal end of nucleocapsids³; AC101's location is supported by our model although we could not identify the density of pp78/83. AC101 has been implicated in actin binding and nucleocapsid movement⁴, supporting its assignment to the nucleocapsid surface.
3. AC104 (VP80) is known to associate at the end of the nucleocapsid⁵ and interact with AC98¹. Deletion of this protein results in nucleocapsids that cannot move from the virogenic stroma⁵, suggesting it is on the outermost surface of the nucleocapsids, which is also in agreement with our capsid model.
4. AC109 was shown to be essential, and deletions/mutations can block nucleocapsid formation and/or result in defective particles^{6,7}.
5. AC142 is an essential nucleocapsid protein and deletions or mutations affect nucleocapsid formation^{8,9}. Partial deletion of AC142 results in non-enveloped, elongated nucleocapsid with no packaged genome⁸. AC109 and AC142 coimmunoprecipitate and form a complex in virus replication¹⁰. All these results support our models and there are broadly interactions between AC109 and AC142.
6. AC144 interacts with AC101² and deletion of AC144 results in amorphous structures containing VP39; nucleocapsids do not form⁸.

As such, the current information regarding the location and role of the AcMNPV structural proteins is supported by our models. Based on this, coupled with the cross-linking mass spec and improved map/model agreement, we are confident in the assignment of the protein subunits in the density map.

We have modified the manuscript to provide additional validation information.

Other comments in no particular order

1) The movies are a very nice addition, and really help with understanding the structure.

A: Thank you for the positive comment.

2) Page 6: The figure for the putative portal needs to be improved. Also, the discussion of the putative portal is somewhat ambiguous. Some of that may be by design, but that section of the manuscript seems somewhat tacked on.

A: Thank you for the suggestions. We have modified the Supplementary figure 2 to show the putative portal structure. We also modified the discussion on the portal.

3) Fig 1: What is the structure of the region between b and c and c and d?

A: In this research, each ODV particle has a helical cylinder with about 220nm or 310nm in length (see FigureS2 and FigureS7). Fig.1 C illustrates the structure one typical portion of whole helical cylinder structure. Therefore, the structures between b and c, c and d are same as that shown in Fig.1C. We modified the figure legend of Fig.1.

4) P13 line 340 -refine > refinement

A: We have change "refine" to "refinement" in the revised version.

Reviewer #3 (Remarks to the Author):

The manuscript "Novel architecture of the baculovirus nucleocapsid revealed by cryo-EM" describes the first high resolution structure of a baculovirus capsid. The authors used a combination of helical reconstruction, single particle analysis, symmetry relaxation, and focused reconstruction to determine structures of the central cylindrical region of the capsid and the two ends. When density quality was good enough, the authors were able to build atomic models directly into their maps. To complete the model and identify additional structural proteins, the authors used alphafold2 to fold various ORFs, and then used their experimental maps as filters to select the correct model. The manuscript is clearly written, with beautiful figures and movies. Some of most interesting findings were unexpected similarities to structural proteins in tailed dsDNA bacteriophages. In particular, the major capsid protein resembles the HK97-fold that all dsDNA phages use as their capsid proteins, and they report a protein at one end that looks a phage portal protein. In addition to the evolutionary relationships such similarities suggest, the overall architecture of the capsid is somewhat unique, providing new insights into the geometry of virus capsids.

A: Thank you for the positive comments.

Given the unexpected similarities to phage proteins, the paper could improve by elaborating on the suggested evolutionary relationships. For example it is difficult to tell how much the capsid protein resembles the HK97 fold. I understand the structural homology is limited, but is it possible to show a superposition of homologous regions/domains? Do the proteins share the same topologies, which might suggest divergent evolution, or do they only position certain secondary structural elements in similar ways (e.g. convergent evolution).

A: Thank you for the suggestions. We have modified supplementary Figure3 such that similar domains are shown with the same colors. It is difficult to infer the evolutionary relationship of these proteins as they show little to no sequence similarity. What we can clearly observe though is that there is structural conservation of key structural elements:

the spine helix, the E-loop and the 4-stranded beta sheet in the A-domain. And while there are some obvious differences in the two structures, these conserved features occurs in relatively similar locations of the sequence and structure. A number of different types of viruses, including herpesviruses, have reported a similar overall architecture, suggesting that these conserved features may provide a modular/adaptable framework for constructing viral capsids. To the more specific point of convergent or divergent evolution, the capsid proteins, at least structurally, seem to share much of the same topology, but can accommodate significant changes to the changes as long as the conserved elements remain intact. It is hard to draw any concrete conclusion on evolution, but this may be a case of divergent evolution.

Additionally, it is difficult to tell how well the T4 portal fits in their density, and it is confusing that they dropped a 12-fold symmetric structure in 7-fold symmetric density. Again, some sort of super-position, or topology diagram would be helpful. The authors might also want to take a look at a recent PNAS paper about archaeal spindle-shaped viruses; the authors of that paper found a protein that seems quite similar to phage portals, and that protein is a heptamer.

A: Thank you for the comments and suggestions. Indeed, describing the portal-like density is a bit complicated and messy. The work on AcMNPV is ongoing and we are focused on pushing the resolution of the AcMNPV map. In particular, we hope to identify the components of the portal complex and cap, as well as the symmetry using both single particle cryoEM and cryogenic electron tomography (cryoET). However, at this point, our work has yet to produce any additional definitive information on these parts of the nucleocapsid.

In the current work, the appearance of the cap and portal-like density were a bit unexpected and our initial observation was that, based on size, shape and location, the portal-like density appeared to be similar to the T4 portal structure, despite having different symmetries. We used UCSF's Chimera to fit the model of the T4 portal to our portal-like density; the cross-correlation was 0.864 and the correlation about mean was 0.2645, suggesting that there is reasonable gross structural similarity at the reported map resolution. We have modified supplementary Figure2 to illustrate the fitting. The attached movie file "portal-fitting-slice.mp4" shows slices along the z axis illustrating the fitting in different areas of the portal density.

Despite the relatively high cross-correlation value, the symmetry of the T4 portal and the symmetry used in our reconstruction are different. In order to determine the symmetry of the apical domain, we collected additional data using cryoET and subtomogram averaging in an effort to analyze the apical domain structure and determine the symmetry (Fig.1 and Fig.2). Briefly, we used the "tomo" function in Relion4 to reconstruct the AcMNPV nucleocapsids without imposing symmetry (Fig 1A); a slice through the apical domain is shown in Fig 1B. Analysis of the symmetry, excluding the outer and inner layers of the nucleocapsid were performed. In Fig 1C, the area represented by the blue box

appears to show at least 12 symmetry peaks and, in Fig. 1D, the density contained in the orange box appears to have at least 19 peaks. In Fig 2, the left panel shows a longitudinal section of the average ODV tomogram, while the right panel shows a transverse section from the same reconstruction. In the right panel, we can observe the symmetry of the outer and inner layer, which have been confirmed by the single particle analyses. While the portal-like density visually appears may have C12 symmetry, we cannot confirm this observation at this time. This work is ongoing and will be the focus of future work.

We would also like to thank the reviewer for directing our attention to the C7 portal complex from the spindle shape virus (Proc Natl Acad Sci U S A. 2022 Aug 2;119(31):e2119439119). Indeed, there may be some similar features. We have tried imposing C14, C7 as well as other symmetries though no improvement to the density (resolvability of domains/subunits) could be seen. Again, this is part of ongoing research and hopefully the subject of future manuscripts.

Fig.1 Cryo-ET of ODV and symmetry analysis.

Fig.2 Views of averaged apical domain

Similarly, the authors sensibly suggest that these viruses may package DNA like phages, i.e. build an empty shell and then pump the DNA into that shell. The ATPases that power packaging are usually easy to identify via bioinformatics. Is there an obvious candidate for the packaging ATPase?

A: There are few papers that mention ATPase related information in baculovirus. Jin J, et al., reported that LEF-4 is a subunit of baculovirus RNA polymerase and has RNA 5'-triphosphatase and ATPase activities; it may function as ATPase in transcribing viral genes¹¹. Another candidate that might be involved in DNA packaging is Ac66¹² which was found to be associated with AcMNPV and HearNPV ODV^{13,14}. However, knockouts of Ac66 do not affect DNA packaging¹⁵. The HAD (haloacid dehalogenase) superfamily includes phosphoesterases, ATPases, phosphonates, etc.,¹⁶; AC98, which is found at the inner layer of base and interacts with the DNA strands, has an HAD domain. We speculate that AC98 may act as an ATPase in DNA packaging in AcMNPV, as it was also reported that knockouts of AC98 result in genome packaging failure¹⁷. However, we still need more experimental evidence to support this possibility, and as such, have not addressed this more fully.

We have modified the text in the discussion to show the reviewer's consideration.

Finally, do the authors have any thoughts about a circular dsDNA would be packaged using this strategy?

A: Thank you for the good questions. Unlike linear DNA viruses where the genome packaging mechanisms have been better elucidated, the mechanisms for packaging circular dsDNA genome are a bit "blurrier". The most accepted model for the circular dsDNA genome packaging was proposed by Dr. M.J.Farse in 1986¹⁸. In this model, the circular genome would be wound and packaged into a preassembled "capsid sheath" through the cap. Here, the "cap" should first recognize the initial packaging site, then function as a motor to push the circular dsDNA genome into the preassembled "capsid sheath". All of these points are still not very clear, and, for many years, no better model has been put forward. We think the model presented by Farse is reasonable, especially when we see a portal-like structure in the single particle cryoEM maps, as well as in aforementioned cryo-ET. Unfortunately, the resolution is not enough for us to make any substantive conclusions; additional experimental evidence and high-resolution structural data are required to unveil more detail in regards to the genome packaging strategy for circular dsDNA.

We have modified the text in the discussion to show the reviewer's consideration.

Is the DNA linearized for packaging? If not, is the channel for the portal large enough to allow passage of a circular dsDNA.

A: According to model proposed by Dr. M.J.Farse and their ultrathin section results, the circular dsDNA genome would be wound as it entered into the capsid. The model proposed two dsDNA strands would go through the cap and the DNA would not be linearized. The narrowest diameter in our study is the portal structure (2nm), which is not large enough to allow passage of two dsDNA strands. In our structure, the genome has already been packaged and the portal-like structure should be “closed” to prevent the genome released. It is possible the portal would enlarge as the genome is packaged or released. Secondly, the resolution for the apical domain, including the portal-like structure, is simply not high enough to draw any additional conclusions. We are engaged in further work to try to resolve the structure of the apical domain.

Additional references

- 1 Wu, W. B. *et al.* Autographa californica Multiple Nucleopolyhedrovirus 38K Is a Novel Nucleocapsid Protein That Interacts with VP1054, VP39, VP80, and Itself. *Journal of Virology* **82**, 12356–12364 (2008). <https://doi.org:10.1128/Jvi.00948-08>
- 2 Braunagel, S. C., Guidry, P. A., Rosas-Acosta, G., Engelking, L. & Summers, M. D. Identification of BV/ODV-C42, an Autographa californica nucleopolyhedrovirus orf101-encoded structural protein detected in infected-cell complexes with ODV-EC27 and p78/83. *J Virol* **75**, 12331–12338 (2001). <https://doi.org:10.1128/JVI.75.24.12331-12338.2001>
- 3 Vialard, J. E. & Richardson, C. D. The 1,629-nucleotide open reading frame located downstream of the Autographa californica nuclear polyhedrosis virus polyhedrin gene encodes a nucleocapsid-associated phosphoprotein. *J Virol* **67**, 5859–5866 (1993). <https://doi.org:10.1128/JVI.67.10.5859-5866.1993>
- 4 Li, K. *et al.* The putative pocket protein binding site of Autographa californica nucleopolyhedrovirus BV/ODV-C42 is required for virus-induced nuclear actin polymerization. *J Virol* **84**, 7857–7868 (2010). <https://doi.org:10.1128/JVI.00174-10>
- 5 Marek, M., Merten, O. W., Galibert, L., Vlak, J. M. & van Oers, M. M. Baculovirus VP80 protein and the F-actin cytoskeleton interact and connect the viral replication factory with the nuclear periphery. *J Virol* **85**, 5350–5362 (2011). <https://doi.org:10.1128/JVI.00035-11>
- 6 Fang, M. G., Nie, Y. C. & Theilmann, D. A. Deletion of the AcMNPV core gene ac109 results in budded virions that are non-infectious. *Virology* **389**, 66–74 (2009). <https://doi.org:10.1016/j.virol.2009.04.003>
- 7 Lin, L. *et al.* ac109 is required for the nucleocapsid assembly of Autographa californica multiple nucleopolyhedrovirus. *Virus Research* **144**, 130–135 (2009). <https://doi.org:10.1016/j.virusres.2009.04.010>
- 8 Vanarsdall, A. L., Pearson, M. N. & Rohrmann, G. F. Characterization of

- baculovirus constructs lacking either the Ac 101, Ac 142, or the Ac 144 open reading frame. *Virology* **367**, 187–195 (2007). <https://doi.org:10.1016/j.virol.2007.05.003>
- 9 McCarthy, C. B., Dai, X., Donly, C. & Theilmann, D. A. Autographa californica multiple nucleopolyhedrovirus ac142, a core gene that is essential for BV production and ODV envelopment. *Virology* **372**, 325–339 (2008). <https://doi.org:10.1016/j.virol.2007.10.019>
- 10 Lehiy, C. J., Wu, W., Berretta, M. F. & Passarelli, A. L. Autographa californica M nucleopolyhedrovirus open reading frame 109 affects infectious budded virus production and nucleocapsid envelopment in the nucleus of cells. *Virology* **435**, 442–452 (2013). <https://doi.org:10.1016/j.virol.2012.10.015>
- 11 Jin, J., Dong, W. & Guarino, L. A. The LEF-4 subunit of baculovirus RNA polymerase has RNA 5'-triphosphatase and ATPase activities. *J Virol* **72**, 10011–10019 (1998). <https://doi.org:10.1128/JVI.72.12.10011-10019.1998>
- 12 Rohrmann, G. F. in *Baculovirus Molecular Biology* (ed 4th) (2019).
- 13 Braunagel, S. C., Russell, W. K., Rosas-Acosta, G., Russell, D. H. & Summers, M. D. Determination of the protein composition of the occlusion-derived virus of Autographa californica nucleopolyhedrovirus. *Proc Natl Acad Sci U S A* **100**, 9797–9802 (2003). <https://doi.org:10.1073/pnas.1733972100>
- 14 Deng, F. *et al.* Proteomics analysis of Helicoverpa armigera single nucleocapsid nucleopolyhedrovirus identified two new occlusion-derived virus-associated proteins, HA44 and HA100. *J Virol* **81**, 9377–9385 (2007). <https://doi.org:10.1128/JVI.00632-07>
- 15 Ke, J. H., Wang, J. W., Deng, R. Q. & Wang, X. Z. Autographa californica multiple nucleopolyhedrovirus ac66 is required for the efficient egress of nucleocapsids from the nucleus, general synthesis of preoccluded virions and occlusion body formation. *Virology* **374**, 421–431 (2008). <https://doi.org:10.1016/j.virol.2007.12.033>
- 16 Burroughs, A. M., Allen, K. N., Dunaway-Mariano, D. & Aravind, L. Evolutionary genomics of the HAD superfamily: understanding the structural adaptations and catalytic diversity in a superfamily of phosphoesterases and allied enzymes. *J Mol Biol* **361**, 1003–1034 (2006). <https://doi.org:10.1016/j.jmb.2006.06.049>
- 17 Wu, W. B. *et al.* Autographa californica multiple nucleopolyhedrovirus nucleocapsid assembly is interrupted upon deletion of the 38K gene. *Journal of Virology* **80**, 11475–11485 (2006). <https://doi.org:10.1128/Jvi.01155-06>
- 18 Fraser, M. J. Ultrastructural Observations of Virion Maturation in Autographa californica Nuclear Polyhedrosis Virus Infected Spodoptera frugiperda Cell Cultures. *Journal of ultrasture and molecular and molecular structure research* 189–195 (1986).

REVIEWER COMMENTS

Reviewer #2 (Remarks to the Author):

The authors have made efforts to address my concerns from the original submission. However, there are still some problems that I have substantial concerns about. The manuscript mentions that experimentally observed crosslinks were either partially present or missing in the model. There is no explanation for the inconsistencies. The authors should include the crosslinking data in the manuscript and provide explanations for disagreements with the experiment and their models. On the other hand, the other supporting data are a nice inclusion. However, more detail about how the observations support their model is needed. A table summarizing the supporting data and how it agrees with their model (distance measurements maybe) is needed.

Minor comment

Page 10, Line 263 – “While the map/model agreement provides a quantitatively analysis” – this sentence needs attention.

Reviewer #3 (Remarks to the Author):

The authors have very thoroughly and thoughtfully addressed all the concerns raised by me. Of course, I cannot be sure how well they addressed other reviewers concerns, but their responses again seemed thorough and thoughtfully considered. Congratulations on a beautiful structure and story!

Point-by-point response to the reviewers' comments

We would like to thank all three reviewers for their original responses and the responses to the latest version of the manuscript. Both reviewer 1 and 3 are satisfied with our response, while reviewer 2 raises some additional concern about the cross-linking/mass spec data. Below, we provide a detailed point-by-point response to each comment. The original comments are quoted in black, and our responses are in blue. The changes we have made in the revised manuscript are in blue.

Reviewer #1:

(No more comments)

Reviewer# 3:

The authors have very thoroughly and thoughtfully addressed all the concerns raised by me. Of course, I cannot be sure how well they addressed other reviewers' concerns, but their responses again seemed thorough and thoughtfully considered. Congratulations on a beautiful structure and story!

A: Thanks a lot

Reviewer #2:

Q1: The authors have made efforts to address my concerns from the original submission.

However, there are still some problems that I have substantial concerns about. The manuscript mentions that experimentally observed crosslinks were either partially present or missing in the model. There is no explanation for the inconsistencies.

A1: Ideally, Cross linking mass spectrometry can directly and experimentally validate the protein/density assignments and assess our model. While the mass spec data does seem to validate some of our assignments, not all head and base proteins could not be validated by the experiment. The potential issues may include: 1) We use bis

(sulfosuccinimidyl) suberate (BS³) as linker, which targets the amine groups in lysine side chains, in addition to having some reactivity with the hydroxyl groups in serine, threonine and tyrosine. BS3 has a spacer length of 11.4 Å and can bridge Cα–Cα distances of 26 to 30Å; cross-linking can only occur if the specific atoms are at the correct distance; 2) There are lot of polyhedrin in the sample which may block the cross-linker of nucleocapsid proteins;

However, the mass spec data does not preclude our model, which is strongly supported by independent biochemical results from other labs and fit to our refined density maps, which have been improved to be ~3.5 Å (The FSC curves of locally refined maps are attached at the end of this file), thus enhancing our validation based on map/model agreement.

This has now been further addressed in the Results (line 261-271 in the re-revised version) and Discussion section (line 426-467 in the re-revised version).

Q2: The authors should include the crosslinking data in the manuscript and provide explanations for disagreements with the experiment and their models.

A2: The results from the cross-linking/mass spec experiment are now provided as Supplementary Table 2. We have added further explanations in the Results (line 261-271 in the re-revised version) and Discussion sections (line 426-467 in the re-revised version).

Q3: On the other hand, the other supporting data are a nice inclusion. However, more detail about how the observations support their model is needed. A table summarizing the supporting data and how it agrees with their model (distance measurements maybe) is needed.

A3: Unfortunately, most of the biochemical results do not address specific amino acids, just focusing on the entire protein when reported. We have now cited the reference for the specific mutations. We have added Supplementary Table 3 to summarize the information from previous biochemical results reported by independent research groups, density map observations, and the cross-linking results.

Q4: Page 10, Line 263 – “While the map/model agreement provides a quantitatively analysis”

– this sentence needs attention.

A4: Thank you, this sentence was clarified.

FSC curves for locally refined map of AC104-1 AC104-2 AC104-3 trimer

FSC curves for locally refined map of Ac109 and Ac142 heterodimer

FSC curves for locally refined map of AC98

FSC curves for locally refined map of AC101-1 □ AC101-2 dimerized domains

FSC curves for locally refined map of AC144-1 □ AC144-2 and the “rings” surrounding them

REVIEWERS' COMMENTS

Reviewer #2 (Remarks to the Author):

The authors have responded to my suggestions. Unfortunately, their mass spectrometry results neither confirm nor deny the novel parts of their models. However, their inclusion of Table 3 combined with the improved resolution of their EM maps help allay my concerns. I still do have some concerns about their models, but I do not wish to delay the publication of their manuscript. I think the EM is of very high quality, and their approach for refining the sub volumes and model building is top notch. While there remain some ambiguities, I support publication of this manuscript.

Point-by-point response to the reviewers' comments

We thank the reviewers for their constructive and insightful feedback on our manuscript. Both reviewer 1 and 3 are satisfied with our previous response, however, reviewer 2 still has some concerns about our models. Below, we provide a detailed point-by-point response to each comment from reviewer 2. The original comments are quoted in black, and our responses are in blue.

The comments of Reviewer #2 (Remarks to the Author):

The authors have responded to my suggestions. Unfortunately, their mass spectrometry results neither confirm nor deny the novel parts of their models. However, their inclusion of Table 3 combined with the improved resolution of their EM maps help allay my concerns. I still do have some concerns about their models, but I do not wish to delay the publication of their manuscript. I think the EM is of very high quality, and their approach for refining the sub volumes and model building is top notch. While there remain some ambiguities, I support publication of this manuscript.

The answers literally to the concerns:

The authors have responded to my suggestions. Unfortunately, their mass spectrometry results neither confirm nor deny the novel parts of their models.

A: We appreciate the reviewer's comment. Yes, we were also not initially satisfied with the MS results. As we have reported in the last revised version, the reasons that the MS results couldn't confirm our models may include: 1, there are too many polyhedrin which blocks or limits cross linking of the capsid proteins and 2, the choice of the linker, BS³, which we think may be not suitable for our case. We have planned to additional follow-up cryoEM, cryoET and cross-linking studies that may provide additional information, but this work will likely be addressed in future publications.

However, their inclusion of Table 3 combined with the improved resolution of their EM maps help allay my concerns.

A: We thank the reviewer for the comment and agree that the additional improvement in resolution and corroboration with other biochemical data does provide further model justification.

I still do have some concerns about their models, but I do not wish to delay the publication of their manuscript.

A: We thank reviewer for their kind and conscientious response. We are continuously pushing the resolution further and the current resolution has allowed us to identify more and more side chain densities, thereby validating our models and approach. This is an ongoing project and we hope to report new structural details in the future.

I think the EM is of very high quality, and their approach for refining the sub volumes and model building is top notch. While there remain some ambiguities, I support publication of this manuscript.

A: We thank reviewer for his support and will do our best to answer the remaining questions/ambiguities in future work.